# Melanoma subpopulations that rapidly escape MAPK pathway inhibition incur DNA damage and rely on stress signalling

Chen Yang 1,2,3,4, Chengzhe Tian 1,2,4, Timothy E. Hoffman 1,2,4, Nicole K. Jacobsen1,2 & Sabrina L. Spencer1,2✉

Despite the increasing number of effective anti-cancer therapies, successful treatment is limited by the development of drug resistance. While the contribution of genetic factors to drug resistance is undeniable, little is known about how drug-sensitive cells first evade drug action to proliferate in drug. Here we track the responses of thousands of single melanoma cells to BRAF inhibitors and show that a subset of cells escapes drug via non-genetic mechanisms within the first three days of treatment. Cells that escape drug rely on ATF4 stress signalling to cycle periodically in drug, experience DNA replication defects leading to DNA damage, and yet out-proliferate other cells over extended treatment. Together, our work reveals just how rapidly melanoma cells can adapt to drug treatment, generating a mutagenesis-prone subpopulation that expands over time.

---

[1] Department of Biochemistry, University of Colorado Boulder, Boulder, CO, USA. [2] BioFrontiers Institute, University of Colorado Boulder, Boulder, CO, USA. [3] Department of Molecular, Cellular, and Developmental Biology, University of Colorado Boulder, Boulder, CO, USA. [4]These authors contributed equally: Chen Yang, Chengzhe Tian, Timothy E. Hoffman. ✉email: sabrina.spencer@colorado.edu

Melanomas driven by the BRAF[V600E] mutation are a widely used model to study drug adaptation and resistance. The BRAF[V600E] mutation causes hyper-active signalling through the RAF–MEK–ERK MAPK pathway[1], which leads to hyper-phosphorylation of the retinoblastoma protein (Rb). Hyper-phosphorylation of Rb engages a positive feedback loop that liberates the transcription factor E2F, which drives cell-cycle entry via transcription of genes that promote progression through G1 and S phases of the cell cycle[2] (Fig. 1a). To block the hyper-proliferation driven by BRAF[V600E], ATP-competitive BRAF inhibitors were developed to treat late-stage melanoma patients harbouring the BRAF[V600E] or BRAF[V600K] mutation. However, despite the immediate, positive clinical response to BRAF inhibitors, resistance typically develops within

**Fig. 1 Proliferation cannot be fully repressed in dabrafenib-treated melanoma cells. a** Schematic diagram of MAPK-dependent cell-cycle entry. **b** Apoptotic cell quantification by flow cytometric analyses of Annexin V-FITC and propidium iodide staining. Quantified replicates of late-apoptotic cells in 1-μM dabrafenib are shown for the indicated time points for five different melanoma cell lines. Error bars: mean ± std of at least two biological replicates, representative of two experimental repeats. **c** A375 cells treated with 1-μM dabrafenib for 0 or 72 h and stained for proliferation markers phospho-Rb and EdU, and with Hoechst to mark nuclei. **d** Probability density of phospho-Rb S807/811 intensity in five melanoma cell lines. Dose–response curves showing the percent pRb+ cells after 96 h of dabrafenib treatment, determined by immunofluorescence quantification. For A375, the untreated 96-h DMSO line falls at 60% (compared with 95% reported elsewhere in this study when cells were plated 24 h before fixation) because 96 h of unfettered growth on the plate results in partial contact inhibition. Error bars: as mean ± std of three replicate wells. **e** Percentage of pRb+ and EdU+ cells in five melanoma cell lines treated for the indicated drug doses and lengths of time; value of % positive cells is noted for the highest dose at 96 h. Error bars: mean ± std of three replicate wells. pRb+ and EdU+ cells were defined by Otsu's thresholding. Source data are provided as a Source Data file.

months[3,4]. Significant research effort has been devoted to understand the origin of resistance to BRAF inhibitors, pinpointing genetic mutations that lead to reactivation of the MAPK or other mitogenic signalling pathways that drive proliferation[5,6]. Further explaining this drug-refractory behaviour, a transient drug-tolerant state has been reported both in pre-clinical models[7–11] and in the clinic[11–13]. On more rapid timescales, reactivation of the MAPK pathway[14–19] as well as activation of other signalling pathways[10,20–23] have been widely reported to be involved in generating or maintaining drug tolerance. Importantly, non-genetic drug tolerance has been linked to the later development of genetic mutations that lead to permanent modes of drug resistance[24,25]. Despite these advances in our understanding, drug resistance and tumour relapse remain major issues in melanoma and many other cancers treated with targeted therapies. Thus, a better understanding of the inception of drug tolerance is critical to eliminate the drug-tolerant population that is hypothesized to serve as a reservoir enabling the development of permanent (genetic) resistance.

In this study, we developed novel experimental and computational methods to track thousands of single melanoma cells continuously over the first 4 days of treatment with the BRAF inhibitor dabrafenib and uncovered a striking degree of heterogeneity in drug response. While most cells initially respond to dabrafenib by entering quiescence, a subpopulation of cells can be seen escaping drug action and re-entering the cell cycle within 3 days of drug treatment. Cells that escape drug rapidly revert to the parental drug-sensitive state upon drug withdrawal, clearly implicating a non-genetic mechanism that enables proliferation in drug. We mapped these drug-escaping cells to a subpopulation of cycling cells visible by single-cell RNA sequencing (scRNA-seq), which revealed a number of pathways uniquely activated in these cells, including the ATF4 stress response. Knockdown of ATF4 in the presence of dabrafenib increased apoptosis and reduced the number of cycling cells by about 50%. Furthermore, cells that cycle in dabrafenib incur increased DNA replication stress and DNA damage, yet outgrow the rest of the population over extended drug treatment. Taken together, our data show that melanoma cells can rapidly rewire to proliferate in the presence of clinical MAPK pathway inhibitors, long before acquisition of genetic mutations.

## Results

**Extensive heterogeneity in the temporal dynamics of single-cell drug responses.** To examine the initial response to dabrafenib, we treated BRAF[V600E] melanoma lines with the BRAF inhibitor dabrafenib and found that MAPK signalling and proliferation were initially repressed but rebounded within 3 days of treatment (Supplementary Fig. 1a). Apoptosis assays detected a large range of cell death across six cell lines, ranging from 94% in SKMEL267C to 17% in A375 to 8% in SKMEL19 cells after 96 h of 1-μM dabrafenib treatment, consistent with previous findings that BRAF inhibitors induce substantial apoptosis in some cell lines, but not in others[6] (Fig. 1b and Supplementary Fig. 1b, c). Cell lines showing minimal apoptosis nevertheless still respond to the drug to varying degrees, visible as a reduction of cell proliferation. Surprisingly, single-cell immunofluorescence identified a dose-dependent subpopulation of residual proliferating cells in WM164, WM278, SKMEL28, A375 and SKMEL19 melanoma lines, even at high doses of drug (Fig. 1c–e).

Does this residual proliferative population arise from a few initial cells and their offspring, or can many cells cycle occasionally in drug? To answer this question, we tracked single-cell proliferation in real time over 5 days using time-lapse imaging of a fluorescent biosensor for CDK2 activity[26–29] (Fig. 2a) coupled with EllipTrack[29], our new cell-tracking pipeline optimized for hard-to-track cancer cells. In the untreated condition, the vast majority of cells cycle rapidly and continuously (Fig. 2b upper panel and Supplementary Fig. 2a; Supplementary Movie 1). Following drug treatment, some cells respond to dabrafenib by entering a quiescent CDK2low state for the remainder of the imaging period (non-escapee), while others initially enter a CDK2low quiescence but later escape drug treatment by building up CDK2 activity to re-enter the cell cycle and divide (escapee) (Fig. 2b lower panel and Supplementary Movie 2; 'Methods'). Clustering of thousands of single-cell CDK2 activity traces in response to 1-μM dabrafenib revealed that 43% of A375 cells and 19% of WM278 cells escape drug at some point during the 4-day treatment (Fig. 2c). Treatment of A375 cells with a very high dose of drug (10-μM dabrafenib) reduced the fraction of escapees to 10%, consistent with a dose-dependent decrease in cell count over time, but did not eliminate the escape behaviour (Supplementary Fig. 2b, c).

In testing characteristics that enable drug escape, we noted that cell-cycle phase at the time of drug addition did not influence the potential to escape, nor did the existence of a small subpopulation of naturally quiescent cells (Supplementary Fig. 2d–f). To determine if pre-existing drug-resistance mutations were at the root of rapid escape, we examined whether escapees could revert to a drug-sensitive state after drug withdrawal. To isolate escapees for such an analysis, we used mCerulean-tagged Geminin[30], a protein that accumulates in S/G2 and is absent during G0/G1, to identify escapees (Geminin+) and non-escapees (Geminin−) (Fig. 2d). Cells were treated with dabrafenib for 72 h and sorted via fluorescence-activated cell sorting (FACS) into Geminin+ and Geminin− populations, given a 24-h drug-free holiday, and then were filmed upon dabrafenib re-treatment. If the ability to escape from dabrafenib results from pre-existing drug-resistance mutations, the proliferation rate of sorted escapees during the second round of dabrafenib treatment should be significantly higher than that of sorted non-escapees and drug-naïve cells (Fig. 2d). Instead, we observed that the proliferation rate was indistinguishable among these three populations (Fig. 2e), indicating that the ability to escape from dabrafenib during the first few days of treatment is not due to pre-existing resistance mutations but rather to a reversible cellular rewiring.

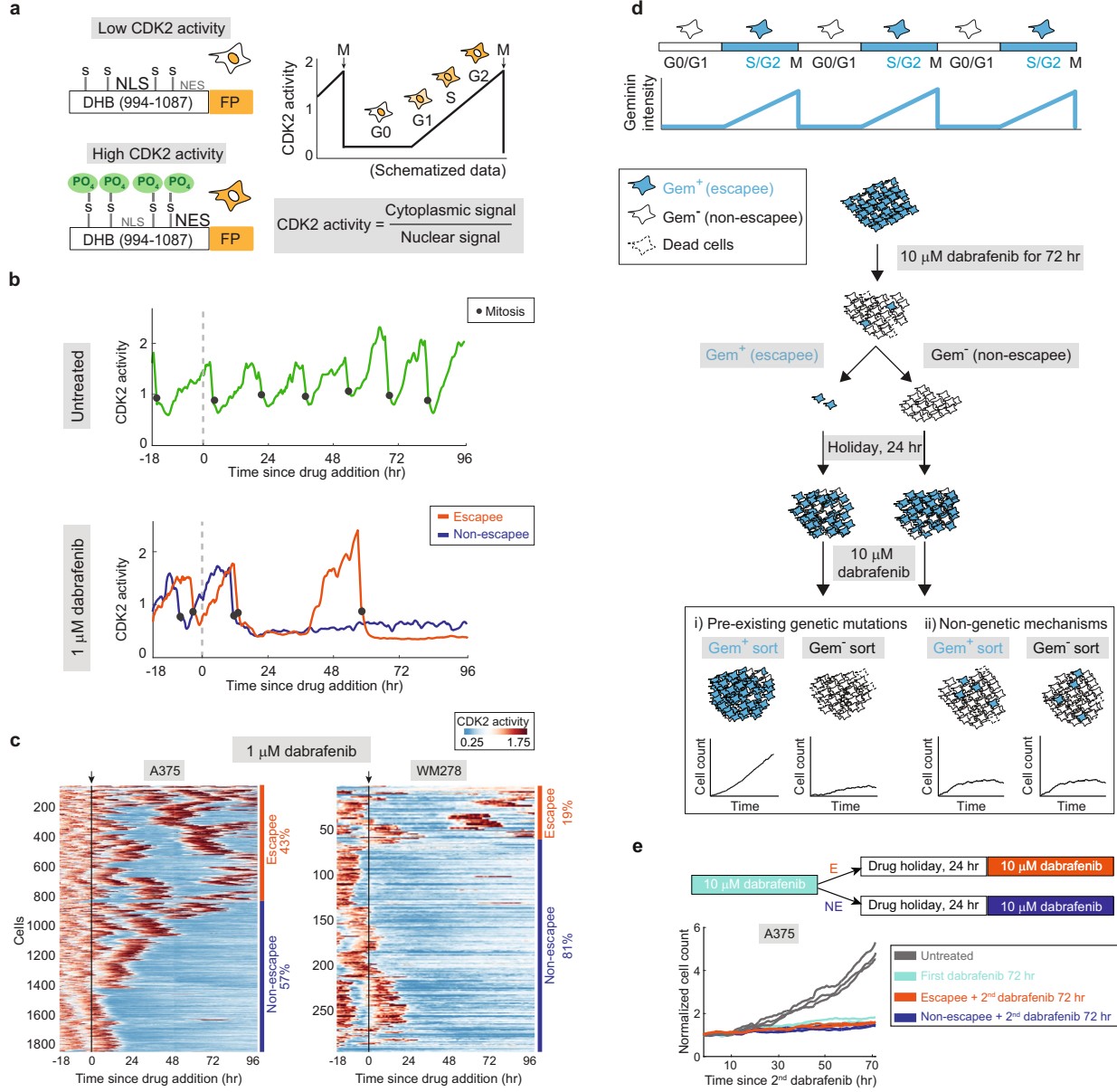

**Fig. 2 A subpopulation of melanoma cells can rapidly and reversibly escape BRAF inhibition. a** Schematic of CDK2 sensor[26–29]. Adapted from refs. [26,27,29], with permission from Elsevier. **b** Representative single-cell traces of CDK2 activity in an untreated A375 cell (upper panel), and a 1-μM dabrafenib-treated escapee and non-escapee (lower panel). **c** Heatmap of single-cell CDK2 activity traces in 1-μM dabrafenib-treated A375 and WM278 cells. Each row represents the CDK2 activity in a single-cell over time according to the colormap. Apoptotic cells (Supplementary Fig. 1c) are not included in the heatmap. The percentages mark the proportion of cells with each behaviour. Arrow and black line mark the time of drug addition. **d** Schematic diagram of the drug holiday experimental setup described in the text. **e** Cell count over time as measured by time-lapse microscopy after a 24-h drug holiday; each condition was measured in triplicate and plotted individually.

**The MAPK pathway is reactivated in escapees**. To ascertain the mechanism underlying the ability of escapees to proliferate in drug, we examined the mRNA expression of ERK downstream targets, *FOSL1*, *ETS1* and *MYC*, and found that their expression levels were higher in escapees (phospho-Rb⁺) than non-escapees (phospho-Rb⁻) (Fig. 3a), indicating that MAPK signalling is increased in escapees relative to non-escapees. Consistently, co-treatment of dabrafenib with trametinib, a clinically used MEK1/2 inhibitor that blocks MAPK pathway reactivation[4,15,17,31], reduced the fraction of escapees to 3% in A375 and 6% in WM278 (Fig. 3b–d and Supplementary Movie 3, 4). We also co-treated cells with dabrafenib and

SHP099, an allosteric inhibitor of SHP2 that suppresses signalling from multiple Receptor Tyrosine Kinase activations[17,19,32,33], which also reduced the fraction of A375 escapees to 3% (Supplementary Fig. 2g, h). Our data suggest that receptor tyrosine kinases are rapidly activated after dabrafenib treatment and contribute to ERK reactivation, consistent with extensive literature documenting a key role for MAPK pathway reactivation upon BRAF inhibition in melanoma[14–19]. However, escapees could not be fully eliminated by dabrafenib/trametinib or dabrafenib/SHP099 combination therapy, suggesting that adaptive mechanisms in addition to MAPK pathway reactivation must exist.

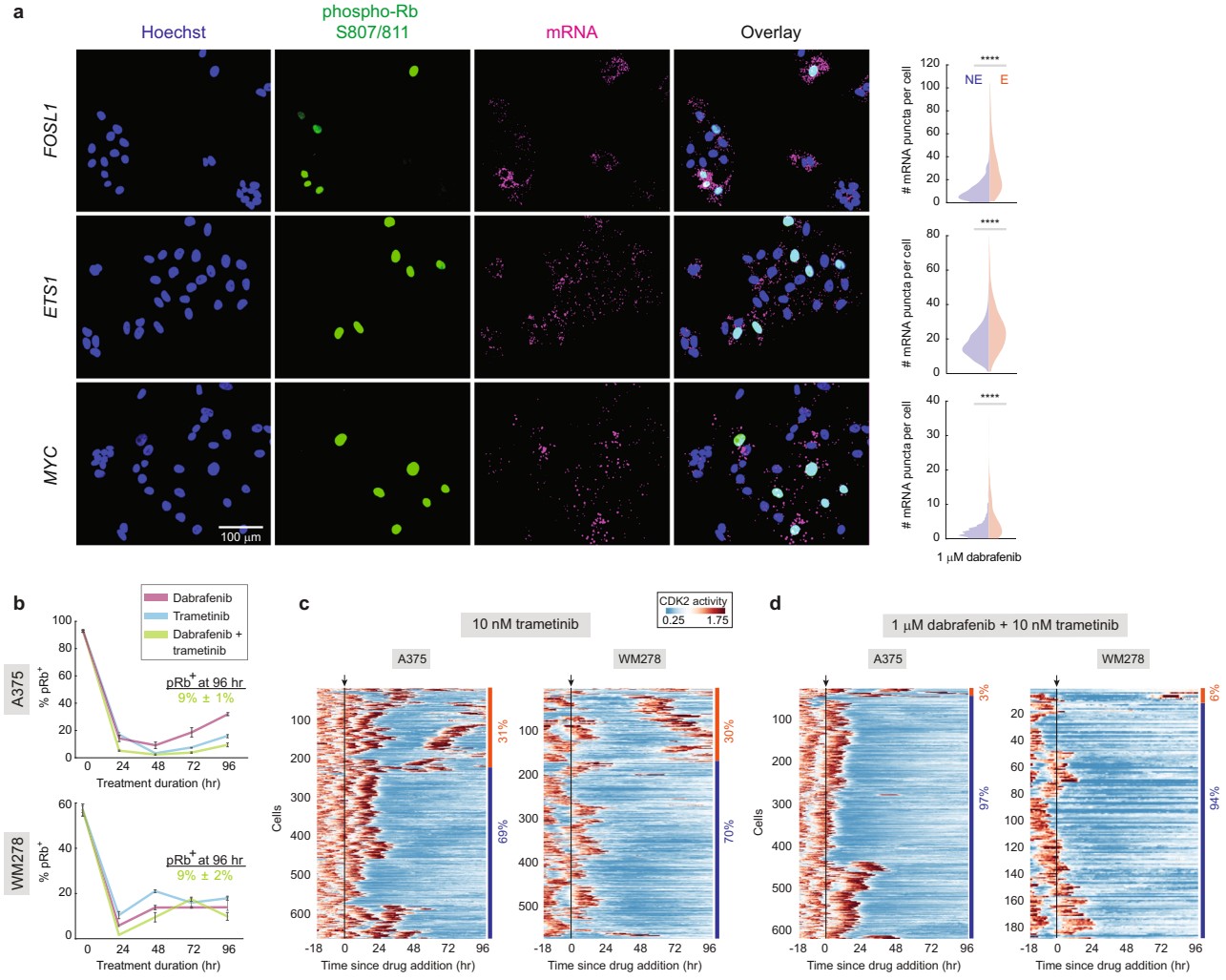

**Fig. 3 The MAPK pathway is reactivated in escapees. a** Representative RNA-FISH images for *FOSL1, ETS1 and MYC* with phospho-Rb (S807/811) and Hoechst staining in A375 cells treated with 1-μM dabrafenib for 72 h. Split violin plots show the number of mRNA puncta in escapees (E) or non-escapees (NE). **b** Quantification of percentage of pRb⁺ cells in A375 and WM278 cells treated for the indicated durations with 1-μM dabrafenib or 10-nM trametinib alone or in combination. The percentage of pRb⁺ cells under the combined treatment at 96 h is noted. Error bars: mean ± std of three replicate wells. **c, d** Heatmap of single-cell CDK2 activity traces in 10-nM trametinib alone or combination of dabrafenib and trametinib, in A375 and WM278 cells. Each row represents the CDK2 activity in a single cell over time according to the colormap. Apoptotic cells are not included in the heatmap. The percentages mark the proportion of cells with each behaviour. Arrow and black line mark the time of drug addition. Source data are provided as a Source Data file.

**Single-cell transcriptomic profile of escapees**. To identify additional adaptive mechanisms, we performed scRNA-seq on A375 cells treated with dabrafenib for 72 h (Supplementary Fig. 3a). We identified escapees by computing the proliferation probability based on expression of 51 cell-cycle genes (Supplementary Data File 1), two of which we validated by RNA fluorescence in situ hybridization (RNA-FISH; Supplementary Fig. 3b, c). Cells with a proliferation probability of one were classified as escapees and cells with probabilities lower than e⁻⁴⁰ as non-escapees (Supplementary Fig. 3d; 'Methods'), which can be visualized on a t-distributed stochastic neighbor embedding (t-SNE) plot. Escapees can then be identified as a small peninsula within the treated population that points downward toward the untreated population (Fig. 4a).

We first assessed gene expression at a population level to relate our data to the 'AXL/MITF phenotype-switch' model derived from bulk RNA-seq data, where cells can either adopt a differentiated AXL^low/MITF^high or a dedifferentiated AXL^high/MITF^low phenotype[34–38]. On average, dabrafenib-treated A375 cells adopt an AXL^low/MITF^high gene-expression state (Supplementary Fig. 4a–c), consistent with observations that melanoma patients treated with MAPK inhibitors initially show an increase in MITF[11]. In contrast, at the single-cell level, the escapee subpopulation exists in an AXL^high/MITF^low state (Supplementary Fig. 4d, e). Dedifferentiation is often mediated by SOX10 loss[20,34,39] and consistently, we saw lower mRNA expression of SOX10 in escapees, although this was not borne out at the protein level (Supplementary Fig. 4a–c). NGFR, another melanoma drug-resistance marker whose expression marks neural crest stem cells[20,35,37], was induced upon drug treatment, although at the protein level, NGFR levels were in fact lower in escapees relative to non-escapees (Supplementary Fig. 4a–c). Thus, while the treated population on average appears to be in an AXL^low/MITF^high state, the escapee subpopulation is in a more dedifferentiated AXL^high/MITF^low state.

To identify new genes and pathways involved in escape from dabrafenib, we derived a list of genes that were differentially expressed in both dabrafenib-treated escapees vs. non-escapees and in dabrafenib-treated escapees vs. untreated proliferating cells. This analysis yielded 40 upregulated genes and 16

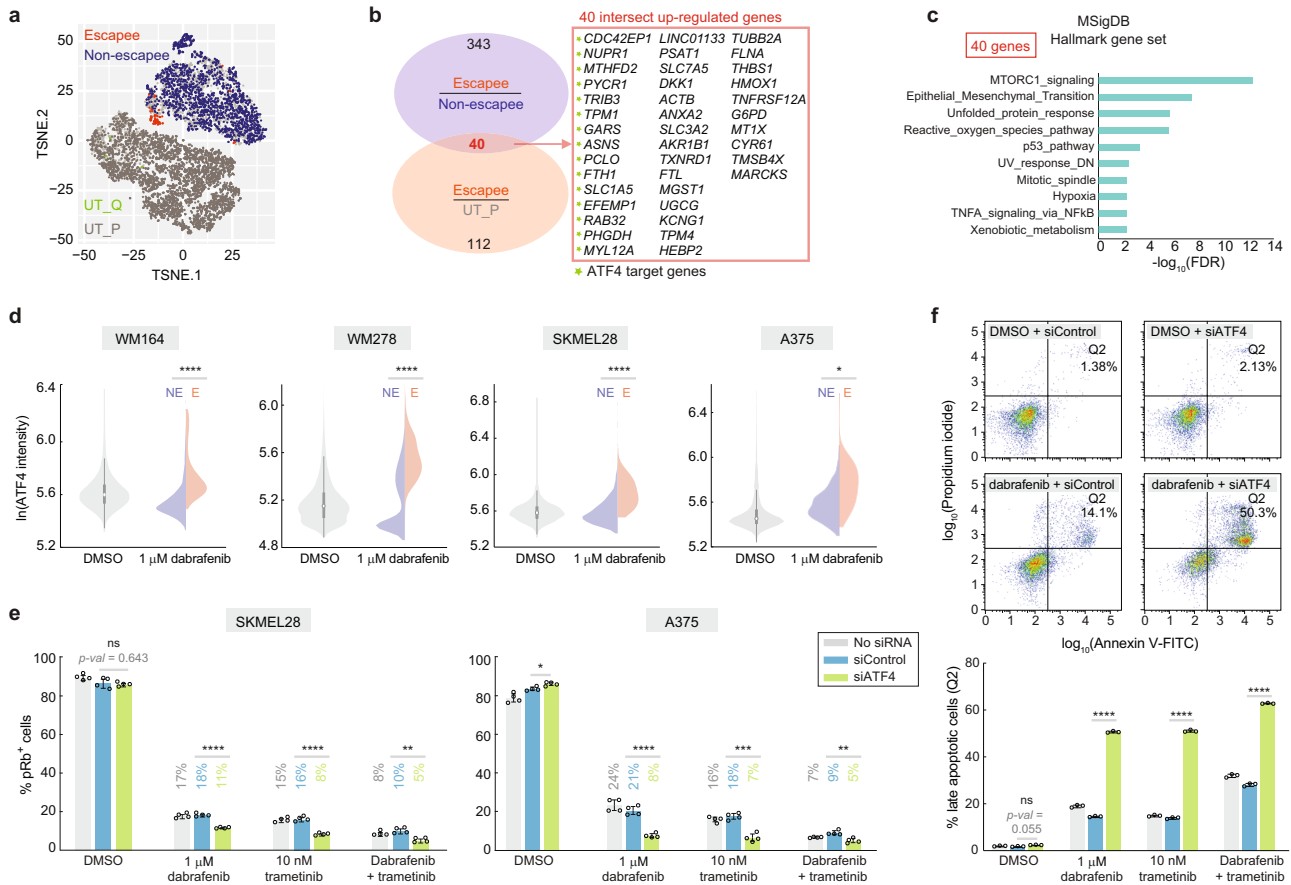

**Fig. 4 scRNA-seq reveals upregulation of ATF4 stress signalling in escapees. a** Co-visualization of untreated (bottom left) and treated (top right) scRNA-seq datasets on a single t-SNE plot, showing escapees in orange and non-escapees in blue (see Supplementary Fig. 3d for classification of escapee and non-escapee). Untreated proliferative cells (UT-P) and quiescent cells (UT-Q) were coloured grey and green, respectively. Escapees can be identified as a small orange peninsula in the treated condition. **b** Venn diagram of differentially expressed genes as described in the text. Genes labelled with a green star are ATF4 target genes. **c** MSigDB hallmark gene set enrichment analysis of the 40 genes using the false-discovery rate cutoff of 0.05. **d** Violin plot showing ATF4 protein levels by immunofluorescence in four melanoma cell lines treated with 1-μM dabrafenib for 72 h. Split violin shows ATF4 levels in non-escapees (NE) and escapees (E) identified by phospho-Rb (S780) co-staining. Each population value is pooled from three replicate wells. **e** The percentage of pRb+ cells in the indicated conditions. Cells were treated for 72 h. Error bars: mean ± std of four replicate wells, representative of two experimental repeats. **f** Percentage of late-apoptotic A375 cells with or without ATF4 knockdown, after a 96-h treatment with the indicated drugs. Error bars: mean ± std of three biological replicates, representative of two experimental repeats. Source data are provided as a Source Data file.

downregulated genes (Fig. 4b and Supplementary Data File 1). Cis-regulatory sequence analysis in iRegulon[40] revealed 15 transcriptional targets of ATF4 among the 40 upregulated genes. ATF4 is induced by the integrated stress response (ISR), which impairs general translation but enhances translation of ATF4, leading to upregulation of a group of stress-responsive genes[41]. Interestingly, ATF4 was reported to maintain an $AXL^{high}$/$MITF^{low}$ phenotype in melanoma[42], consistent with our findings. Pathway enrichment analysis with the Hallmark Gene Set[43] identified mTORC1 signalling as enriched in escapees, in addition to other stress–response signatures such as unfolded protein response (a component of the ISR), oxidative stress, and p53-dependent pathways (Fig. 4c). In addition, the epithelial–mesenchymal transition pathway was enriched, consistent with escapees having a mesenchymal-like dedifferentiated gene signature.

Activation of mTORC1 and ATF4 pathways in escape from BRAF inhibition illuminates potential new targets to extinguish the escapee population. Indeed, co-treatment with dabrafenib and the mTORC1 inhibitor rapamycin blocked the rebound of escapees normally seen with dabrafenib alone (Supplementary Fig. 5a). In addition, ATF4 mRNA and protein were upregulated

upon dabrafenib treatment, with protein levels particularly high in escapees, in four cell lines tested (Fig. 4d and Supplementary Fig. 5b). Dabrafenib-mediated ATF4 upregulation could be reduced upon co-treatment with rapamycin, suggesting that mTORC1 and ATF4 activities may be coupled (Supplementary Fig. 5c). If ATF4 activation enables escape from dabrafenib, then its depletion should reduce the percentage of escapees. Indeed, siRNA knockdown of ATF4 reduced escapees by 60% and 40% relative to control siRNA under dabrafenib treatment condition in A375 and SKMEL28 cells, respectively (Fig. 4e). Importantly, knockdown of ATF4 in cells treated with trametinib alone or dabrafenib plus trametinib also significantly reduced the escapee population, indicating a role for ATF4 in escape from drugs beyond dabrafenib alone (Fig. 4e). Measurement of apoptosis revealed a striking increase in cell death after ATF4 knockdown in A375 cells treated with dabrafenib, trametinib or the combination, suggesting that ATF4 stress signalling promotes cell survival under the targeted therapies tested (Fig. 4f).

As the top hit among ATF4 target genes, CDC42EP1, a member of the Rho GTPase family[44], represents a candidate that may enable escape. Another induced ATF4 target gene of interest is RAB32, belonging to a family of Ras-related GTPases[45]. The

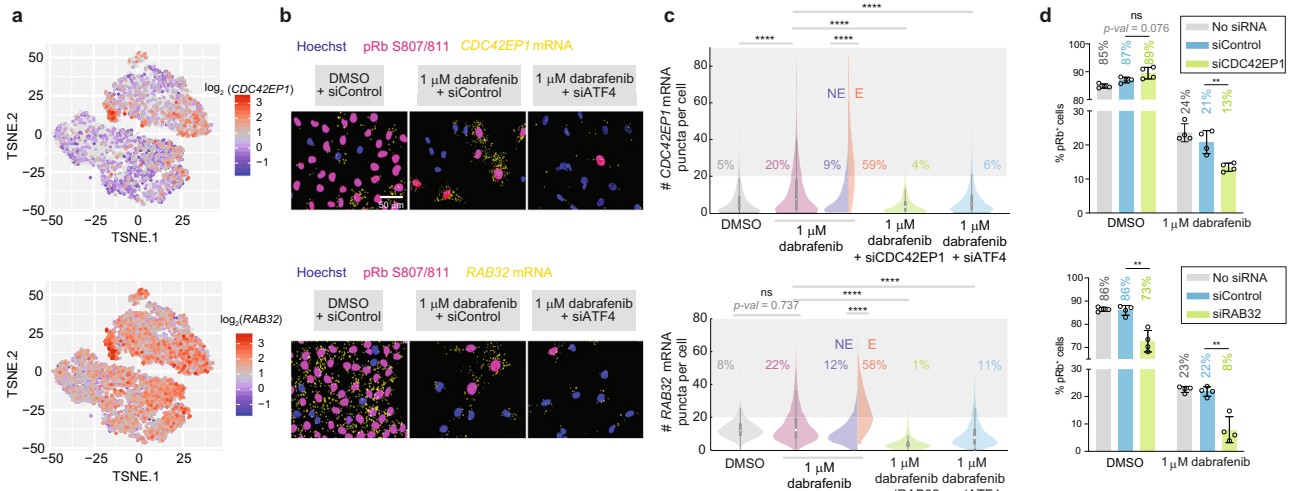

**Fig. 5 Individual ATF4 target genes are involved in escape from dabrafenib-induced quiescence. a** Visualization of single-cell *CDC42EP1* and *RAB32* mRNA expression levels on the combined t-SNE plot showing increased expression in the peninsula containing escapees. **b** Representative RNA-FISH images for *CDC42EP1* or *RAB32* with phospho-Rb (S807/811) and Hoechst staining. **c** Violin plot showing the number of mRNA puncta for *CDC42EP1* or *RAB32* in A375 cells treated under the indicated conditions for 72 h. The percentage of cells with >20 mRNA puncta is indicated on the plot. Each population value is pooled from two replicate wells. **d** The percentage of pRb+ cells in the indicated conditions. A375 cells were treated for 72 h. Error bars: mean ± std of four biological replicates, representative of two experimental repeats. Source data are provided as a Source Data file.

mRNA expression of both genes was dramatically enriched in escapees upon dabrafenib treatment (Fig. 5a, b) and markedly decreased after *ATF4* knockdown, confirming their ATF4 target status (Fig. 5b, c). Knockdown of *CDC42EP1* or *RAB32* in dabrafenib-treated cells significantly reduced the percentage of escapees compared to siControl (Fig. 5d). Similar results were obtained for these two genes in WM278 cells (Supplementary Fig. 5d–f). *LINC01133*, a lncRNA of unknown function and the top hit among non-ATF4 target genes, was also significantly enriched in escapees (Supplementary Fig. 5g, h) and contributed to dabrafenib-mediated escape (Supplementary Fig. 5i), indicating that a fraction of escapees relies on ATF4-independent mechanisms to proliferate in drug.

**Genes upregulated in escapees show clinical significance**. To probe the translational relevance of our findings, we tested for the existence of escapees in two short-term ex vivo cultures of BRAF$^{V600E}$ melanoma patient biopsies obtained prior to treatment (MB4562 and MB3883). Dose–response curves revealed residual cycling cells even at the maximal dose of dabrafenib (Fig. 6a), indicating the presence of escapees in these patient samples. Using the IC50 for each patient sample, we observed a decrease in cycling cells at 4 days of treatment followed by a significant increase at 7 days of treatment, similar to the rebound in proliferation observed in commercial lines (Fig. 6b). We then measured both ATF4 and phospho-S6, a marker for mTORC1 activity, in the more drug-sensitive patient sample MB3883, and found that both signals steadily increased throughout a week of treatment, with significant enrichment in escapees (Fig. 6c, d). Consistent with the observed ATF4 induction, *CDC42EP1* and *RAB32* mRNA levels were also induced in dabrafenib-treated MB3883 cells with significant enrichment in escapees relative to non-escapees (Fig. 6e, f).

To determine the long-term clinical relevance of these results, we assessed our list of 40 genes uniquely upregulated in A375 escapees for impact on melanoma patient survival using The Cancer Genomics Atlas (TCGA). Five out of the 40 genes were associated negatively with patient survival, including *CDC42EP1*, *RAB32* and several other ATF4 and mTORC1-associated targets (Fig. 6g, Supplementary Fig. 6 and Supplementary Data File 2),

while none associated positively, representing a strikingly high proportion of the 40 genes (permutation analysis, $p = 0.004$). Thus, genes uniquely upregulated in A375 escapees after just days of dabrafenib treatment show negative long-term correlation with patient survival assessed over decades.

Do these findings pertain only to dabrafenib treatment, or do they extend to other MAPK pathway inhibitors? We treated two commercial lines (A375 and WM278) and the two ex vivo patient biopsies (MB4562 and MB3883) with several BRAF inhibitors (dabrafenib, vemurafenib, PLX 8394) and a MEK1/2 inhibitor (trametinib). Cycling cells could not be completely eliminated by any of the treatments, even at high drug doses (Fig. 6h). In addition, ATF4, *CDC42EP1* and *RAB32* levels were specifically enriched in escapees in every treatment condition (Supplementary Fig. 7a, b).

**Escapees are prone to DNA damage, yet out-proliferate non-escapees over extended treatment**. Could escapees be the seed population driving eventual acquisition of drug-resistance mutations? For this to be the case, escapees would have to be both prone to mutagenesis and out-proliferate non-escapees. Indeed, A375 and MB3883 escapees showed increased γ-H2AX staining and increased DNA double-strand breaks relative to non-escapees (Fig. 7a, b and Supplementary Fig. 8a, b). Furthermore, dabrafenib-treated EdU+ cells reach a lower EdU intensity maximum compared with untreated cycling cells (Supplementary Fig. 8c), suggesting a reduced DNA synthesis rate. Suspecting aberrant DNA replication in escapees, we stained cells for FANCD2, a protein that localizes to stalled replication forks[46]. FANCD2 staining was elevated in drug-treated cells and was particularly high in EdU+ escapees undergoing DNA replication (Fig. 7c). Another potential cause of reduced DNA synthesis rate is the under-licensing of origins of replication prior to the start of S phase. We therefore measured the amount of the licensing factor MCM2 bound to chromatin in single cells[47] and found marked under-licensing after dabrafenib and trametinib treatment (Fig. 7d and Supplementary Fig. 8d–f). Thus, escapees experience dysregulated licensing and heightened DNA replication stress, which are known to cause genomic instability in

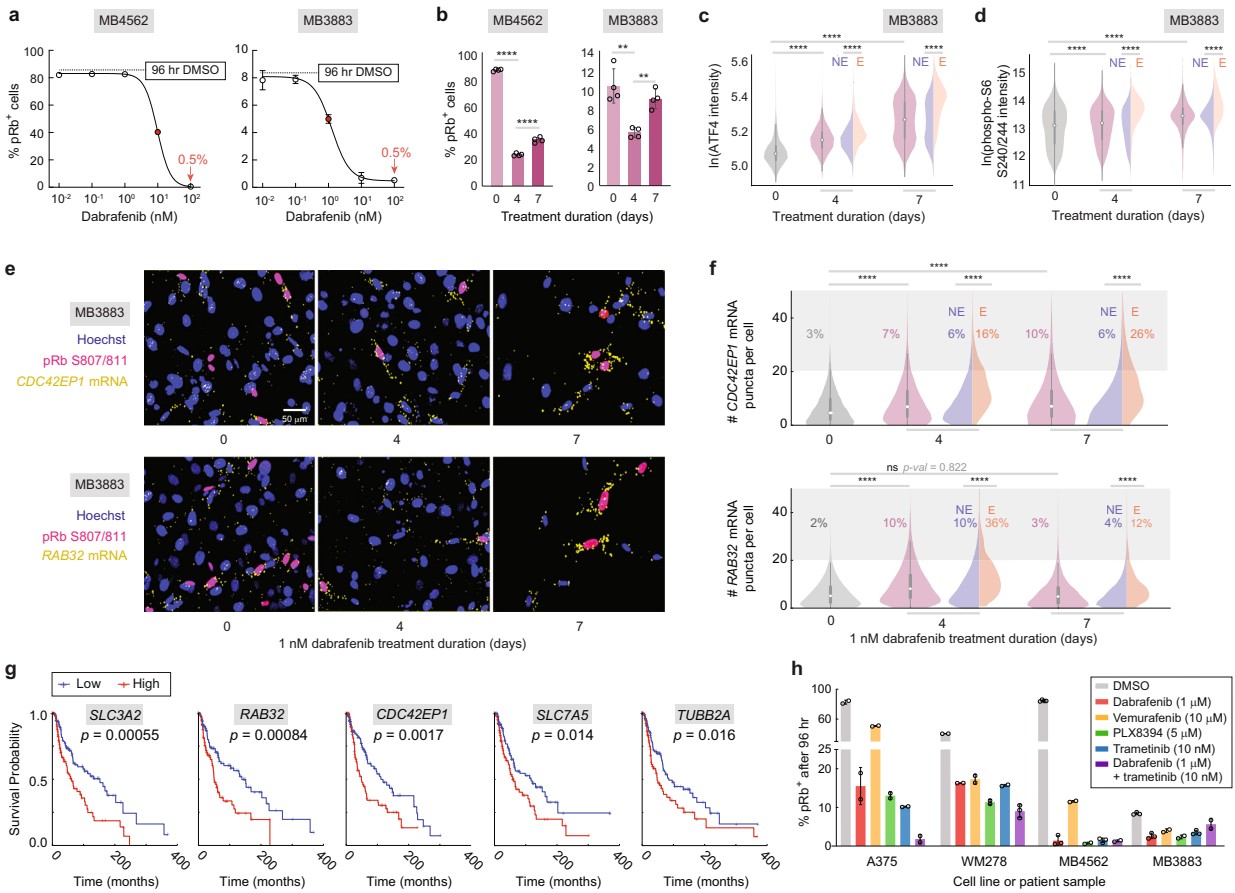

**Fig. 6 Existence of escapees and upregulation of ATF4 target genes in clinical samples. a** Dose–response curves at 96 h of treatment in two ex vivo patient cultures, noting a residual 0.5% of pRb+ cells at the highest dose. Approximate IC50 values are displayed as red dots. Error bars: mean ± std of three replicate wells. **b** Percentage of pRb+ cells in the two patient samples treated with IC50 dose of dabrafenib for 4 or 7 days. Error bars: mean ± std of four replicate wells, representative of two experimental repeats. **c, d** Violin plots showing ATF4 and phospho-S6 (S240/244) levels by immunofluorescence in MB3883 patient cells treated with 1-nM dabrafenib for 0, 4 or 7 days. Escapees are identified by phospho-Rb (S780) or EdU co-staining for ATF4 or phospho-S6, respectively. Each population value is pooled from six replicate wells. **e** RNA-FISH images for *CDC42EP1* and *RAB32* with pRb (S807/811) and Hoechst staining, for MB3883 cells cultured in 1-nM dabrafenib for 0, 4 or 7 days. **f** Violin plots showing the number of *CDC42EP1* or *RAB32* mRNA puncta in MB3883 cells. The percentage of cells with >20 mRNA puncta is indicated on the plot. Each population value is pooled from two replicate wells. **g** Melanoma patient survival curves for 5 of 40 genes upregulated in escapees. *p* value: log-rank test. **h** Percentage of pRb+ cells in A375, WM278 and two ex vivo patient cultures treated with high doses of dabrafenib, vemurafenib, PLX8394, trametinib or dabrafenib plus trametinib for 4 days. Error bars: mean ± std of at least two replicate wells, representative of two experimental repeats. Source data are provided as a Source Data file.

cancer cells[48]. Together, these data demonstrate that cells cycling in the presence of dabrafenib are prone to DNA damage.

To determine whether escapees out-proliferate non-escapees in the longer term, we imaged and tracked dabrafenib-treated A375 cells over 12 days. Cells were classified as escapees or non-escapees based on behaviour in the first 96 h, and their proliferative activity was examined during the remaining days. Non-escapees rarely re-entered the cell cycle during the final 7 or 8 days and had a median of one mitosis during that period. By contrast, escapees cycled significantly more frequently than non-escapees, having a median of two or three mitoses during the final 7 or 8 days, respectively, with long quiescence periods in between (Fig. 7e and Supplementary Fig. 9). To further test whether escapees outgrow non-escapees over an even longer time period, we sorted escapees and non-escapees after 72 h of 1-μM dabrafenib treatment via FACS directly into media containing 1-μM dabrafenib and imaged the separate populations periodically over 1 month in drug. Consistent with the live-cell imaging result, sorted escapees out-compete non-escapees over the 1-month observation time (Fig. 7f; cf. Fig. 2e where cells were given

a 24-h drug holiday after sorting that allowed them to reset to the parental state). Thus, despite incurring DNA damage, escapees out-proliferate non-escapees, suggesting that they may dominate the population in the long term.

## Discussion

A cell's ability to reprogramme into distinct operational states without genetic changes is a poorly understood but critical adaptive process that may explain many instances of drug resistance and tumour relapse. While much of the field of cancer biology has focused on genetic mutations driving drug resistance, the signalling plasticity involved in non-genetic drug adaptation is only now beginning to be appreciated. As part of our effort to tackle this question, we developed a high-throughput, long-term cell-tracking pipeline called EllipTrack[29] to monitor adaptation to drug in single cells in real time. Although some cells die in response to dabrafenib, the majority of cells studied here survive. We therefore avoided the term 'persister' that typically refers to a few rare surviving cells, and instead use the terms escapee/non-

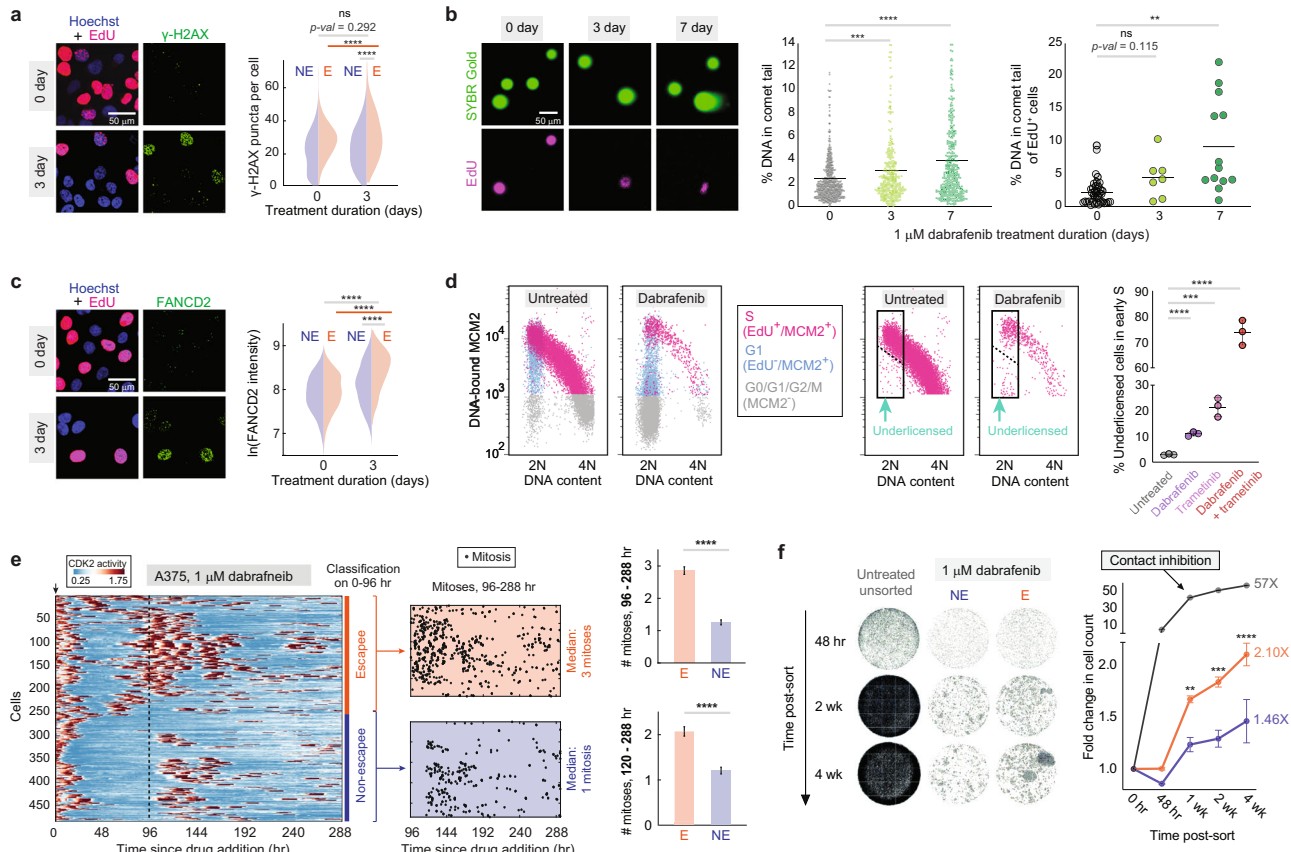

**Fig. 7 Escapees are prone to DNA damage and outgrow non-escapees over extended drug treatment. a** Images of A375 cells treated with 1-μM dabrafenib for 72 h stained for EdU and γ-H2AX. Quantification of γ-H2AX puncta for escapees (E) and non-escapees (NE) is plotted as split violins. $n = 2$ biologically independent experiments. **b** Neutral comet assay in dabrafenib-treated A375 cells. Gels were co-stained for EdU incorporation. Plots indicate percent tail intensity over each entire comet, with mean values displayed as a horizontal line. Left panel: 0, 3 and 7 days, $n = 708, 375$ and $394$ cells. Right panel: 0, 3 and 7 days, $n = 40, 7$ and $13$ cells. **c** Images of A375 cells treated with 1-μM dabrafenib for 72 h stained for EdU and FANCD2. Quantification of FANCD2 intensity for escapees and non-escapees is plotted as split violins. $n = 2$ biologically independent experiments. **d** Flow cytometric analysis of DNA replication licensing determined by chromatin-bound MCM2 in A375 cells treated with BRAF and/or MEK inhibitors. Cells appearing under the dashed line show reduced MCM2 loading at the start of S phase. Right-most plot shows percent under-licensed cells relative to all early S phase cells; mean ± std of three replicate samples. **e** CDK2 activity heatmap for 482 dabrafenib-treated single cells tracked over 12 days. Cell behaviour was classified based on the first 96 h, and subsequent mitosis events are plotted as black dots. Bar plot quantifies the average number of mitoses per cell in the final 7 or 8 days of filming. Error bars: mean ± SEM. $n$ (E; NE) = 248; 234 cells. **f** Escapees (Gem+) and non-escapees (Gem−) cells were separated by FACS after a 72 h of 1-μM dabrafenib treatment and were then continuously cultured in 1-μM dabrafenib for 4 weeks. Representative wells for each condition at different time points are shown. Normalized cell count for each well was quantified and plotted for each condition; mean ± SEM of seven replicate wells. The $p$ value summary represents the comparison between E and NE replicates by unpaired $t$-test. Source data are provided as a Source Data file.

escapee to reflect the proliferation/quiescence effects observed. Within the surviving population, we identified a subpopulation of melanoma cells that were initially responsive to dabrafenib and entered a CDK2$^{low}$ quiescent state, but later re-entered the cell cycle and divided occasionally in the presence of drug. Using FACS to isolate escapees, we showed that the rise of escapees is not due to a pre-existing drug-resistant subpopulation, but rather to a transient, reversible state. Using scRNA-seq, we identified a role for ATF4 in enabling the escapee phenotype, which we validated in several cell lines as well as in melanoma patient samples. Finally, we show that escapees incur DNA damage while cycling in the presence of dabrafenib, yet also outgrow non-escapees over time.

While we used BRAF$^{V600E}$-driven melanoma cells treated with dabrafenib as a model to study drug resistance, our findings may be broadly applicable to other cancers treated with MAPK pathway inhibitors. Indeed, none of the clinically approved MAPK pathway inhibitors tested here alone at high doses or in

combination were able to successfully block proliferation in 100% of cells. This result was consistent for all of the melanoma cell lines analyzed, suggesting that the escape phenotype is neither drug-specific nor cell-line specific. Our results show that cells can readily activate bypass pathways to re-enter the cell cycle. Indeed, activation of the ATF4 pathway represents an evolutionarily conserved general stress–response that may function in adaptation to other clinical MAPK inhibitors not examined here. ATF4 stress signalling is reported to have dual effects under persistent stress: pro-survival[49–54] and pro-apoptotic[53,55]. Here, we report the involvement of the ATF4 pathway in escape from trametinib and dabrafenib, demonstrated by an increase in apoptosis and a decrease in the number of escapees upon ATF4 knockdown, consistent with a pro-survival role. Additional work is needed to determine how widespread the phenomena of rapid drug escape and reliance on ATF4 are.

Cancer drug resistance is often explained by a Darwinian selection scheme where a rare subpopulation with pre-existing

**Fig. 8 Model describing how rapid, non-genetic drug adaptation can lead to bona fide drug resistance.** Dabrafenib-treated melanoma cells initially use non-genetic mechanisms, such as ATF4 stress signalling, to adapt to and escape from drug-induced quiescence. These early drug-adapted escapees incur DNA damage while cycling in drug, yet out-proliferate non-escapees over extended drug treatment. Thus, escapees are more likely than non-escapees to acquire the genetic mutations that lead to permanent drug resistance.

genetic or epigenetic modifications[9,20,56] gains a fitness advantage upon drug exposure and eventually dominates the population. However, some recent studies also identify a Lamarckian induction scheme where cells acquire therapeutic resistance by drug-induced transcriptional reprogramming[57]. Here, we find that escapees rapidly revert to parental drug sensitivity within a 24-h drug holiday, arguing against pre-existing genetic or epigenetic differences in escapees. Moreover, the percentage of escapees in the cell population is highly dependent on the dose of dabrafenib, suggesting that even if a rare drug-resistant subpopulation existed, this subpopulation is unlikely to explain the majority of escapees. Cooperative resistance could also play a role in drug escape, wherein one cell can release stimulatory factors that promote the proliferation of nearby cells[58,59]. However, analysis of spatial correlations in our imaging datasets found no evidence linking escapees to cell-cycle entry of neighbouring cells, although we do not rule out the possibility of an effect during extended treatment or in in vivo settings. Together, these results suggest that the escapee phenotype is rapidly induced, short-lived and reversible, and thus more Lamarckian than Darwinian.

The reversibility of this cell state therefore raises the question of how escapees impact drug resistance in the long term. First, a previous study showed that the dedifferentiated melanoma state can be stabilized through epigenetic reprogramming[20]. Therefore, descendants of escapees are likely to remain dedifferentiated, and tumour relapse might be controlled by these descendants. In support of these speculations, the majority of relapsed tumours were found to have increased *AXL* and reduced *MITF* expression[11,60,61], similar to the escapee population observed here. Second, interrogation of TCGA data showed that a strikingly high percentage of genes (5 of 40) upregulated in A375 escapees at 72 h correlated negatively with patient survival assessed over decades while none correlated positively. Third, it is widely believed that the vast majority of mutations arise during the S phase of the cell cycle[62]. Here, we showed that escapees incur increased DNA replication stress and DNA damage relative to non-escapees. These cells are therefore more prone to mutagenesis and consequently more likely to acquire drug-resistance mutations, consistent with recent work identifying drug-induced mutagenesis in PDX and clinical samples treated with EGFR or BRAF inhibitors[63,64]. Two additional studies from 2016 showed that drug-tolerant cells could give rise to a drug-resistant population in the long term[24,25]. Because escapees incur increased DNA damage and also outgrow non-escapees due to more frequent cell division, we speculate that early escapees from drug treatment could have long-term effects on tumour progression. Development of a combined microscopy and barcode-based lineage tracing strategy would be required to establish a firm connection.

In summary, we discovered that a subset of melanoma cells can rapidly adapt to drug treatment and proliferate via activation of additional signalling pathways. Because escapees are prone to DNA damage and yet out-proliferate non-escapees, they may represent a seed population enabling permanent (genetic) drug resistance (Fig. 8). Since none of the clinically approved MAPK pathway inhibitors tested here alone or in combination successfully block proliferation in 100% of commercial cells or primary patient samples, these findings could have broad applicability, implying that non-genetic escape from targeted therapies may be more common than currently appreciated. New drug combinations targeting mTORC1, ATF4 or the adaptation state more broadly, could forestall drug resistance and tumour relapse.

## Methods

**Cell culture and cell line generation**. The A375 melanoma cell line (#CRL-1619) was purchased from American Type Culture Collection (ATCC). A375 cells were cultured at 37 °C with 5% CO$_2$ in DMEM (Thermo Fisher, #12800-082) supplemented with 10% FBS, 1.5-g/L sodium bicarbonate (Fisher Chemical, #S233-500) and 1X penicillin/streptomycin. The WM164 and WM278 cell lines were obtained from Dr Natalie Ahn (University of Colorado Boulder). The SKMEL19, SKMEL28 and SKMEL267C cell lines were obtained from Dr Neal Rosen (Memorial Sloan Kettering Cancer Center). WM164, WM278, SKMEL19, SKMEL28 and SKMEL267C cells were maintained at 37 °C with 5% CO$_2$ in RPMI1640 (Thermo Fisher, #22400-089) supplemented with 10% FBS, 1X Glutamax, 1X sodium pyruvate (Thermo Fisher, #11360-070) and 1X penicillin/streptomycin. The A375 and WM278 lines used for time-lapse imaging experiments were submitted for short tandem repeat profiling and were authenticated as exact matches to A375 and WM278, respectively. MB3883 and MB4562 cells were obtained from the Cutaneous Oncology Melanoma Bank at the University of Colorado and were maintained in the same conditions as WM164 cells. At the time the patient cells were received, the MB3883 line had been passed through PDX models, whereas the MB4562 line was directly cultured following initial patient biopsy. A375 and WM278 cells were transduced with H2B-mIFP and DHB-mCherry lentivirus or with H2B-mCherry and mCerulean-Geminin lentivirus as described previously[26]. Cells stably expressing these sensors were isolated by two rounds of FACS.

**Small molecules**. Drugs used in this study are: dabrafenib (Selleckchem, #S2807), trametinib (Selleckchem, #2673), SHP099 (Selleckchem, #S8278), vemurafenib (Selleckchem, #S1267), PLX8394 (MedChemExpress, HY-18972), rapamycin (Selleckchem, #S1039) and etoposide (Selleckchem, #S1225).

**RNA-FISH and immunofluorescence**. Cells were seeded on a glass-bottom 96-well plate coated with collagen 24 h prior to drug treatment. Cells were fixed with 4% paraformaldehyde, and when applicable, were processed for RNA-FISH analysis according to the manufacturer's protocol (ViewRNA ISH Cell Assay Kit) (Thermo Fisher, #QVC0001). mRNA probes were hybridized at 40 °C for 3 h, followed by standard amplification and fluorescent labelling steps also at 40 °C. Probes used in this study are ViewRNA Type 6 probes from Thermo Fisher: *FOSL1* (VA6-3168888-VCP), *ETS1* (VA6-3169957-VC), *MYC* (VA6-10461-VC), *CDC42EP1* (VA6-3170107-VC), *RAB32* (VA6-3175871-VC), *CCNA2* (VA6-15304-VC), *CCNB1* (VA6-16942-VC) and *LINC01133* (VA6-20432-VC). For quantification of individual mRNA puncta, cells were stained with total protein dye, CF 568 succinimidyl ester (1:100,000) (Millipore sigma, #SCJ4600027), to create a whole-cell mask for segmentation. FISH images were taken on PerkinElmer Opera Phenix high-content screening system with a 20×1.0 numerical aperture (NA) water objective.

For immunofluorescence, standard protocols were used: following blocking, primary antibodies were incubated overnight at 4 °C, and secondary antibodies

were incubated for 2 h at room temperature. Immunofluorescence stains without RNA-FISH were imaged on a Nikon Ti-E using a 10×0.45 NA objective.

**Live-cell imaging**. Cells were seeded on a glass-bottom 96-well plate coated with collagen 24 h prior to the start of imaging. Movie images were taken on a Nikon Ti-E using a 10×0.45 NA objective with appropriate filter sets at a frequency of 15 min per frame. Cells were maintained in a humidified incubation chamber at 37 °C with 5% $CO_2$. Cells were imaged in phenol red-free full-growth media (Corning, #90-013-PB) for 18 h before treatment; the movie was then paused for drug addition and imaging continued for another 48 h at which point the drug was refreshed; imaging then continued for another 48 h. The drug refreshment was performed by exchanging half of the total media in each well to avoid cell loss during pipetting.

**Definition of escapees**. In time-lapse microscopy experiments, escapees are defined as cells that re-enter the cell cycle after spending any amount of time in a drug-induced CDK2$^{low}$ quiescence before re-entering the cell cycle. Non-escapees are defined as cells that enter into and remain in a drug-induced CDK2$^{low}$ state until the end of the imaging period.

In fixed-cell experiments, escapees are defined as cells that are in the cell cycle after 2 or more days of drug treatment. A cell in the cell cycle can be detected by immunofluorescence staining for phospho-Rb S807/811 or phospho-Rb S780, by EdU incorporation, by detection of Geminin-mCerulean signal, or by expression of cell-cycle genes in scRNA-seq (see 'Calculation of proliferation probability').

**FACS on escapees and non-escapees**. A375 cells expressing H2B-mCherry and mCerulean-Geminin were treated with 10-μM dabrafenib for 72 h before sorting on an Aria Fusion FACS machine. Untreated cells were used to choose the Geminin signal cutoff for Geminin$^+$ (escapees) and Geminin$^-$ (non-escapees). For the experiment in Fig. 2d–e, drug-treated cells were directly sorted into full-growth media for 24 h before the second round of drug treatment and the start of imaging. The subsequent imaging conditions followed the live-cell imaging protocol.

For the experiment in Fig. 7f, cells were constantly maintained in drug throughout the sort at 72 h (including during trypsinization), to prevent the escapees from reverting back to their parental state upon replating. Drug-treated cells were directly sorted into full-growth media containing 1-μM dabrafenib and imaged periodically for 4 weeks (96-well plate format; exactly 2000 cells per well for each condition). The control condition represents untreated cells seeded into untreated media. Nuclei segmentation was based on the H2B-mCherry signal, and cell count was collected via imaging and automated analysis on a PerkinElmer Opera Phenix (10X objective) at 48 h, 1-, 2- and 4-week time points. Cell counts were normalized to the 2000 cells sorted into each well. Drug media was refreshed every 3.5 days throughout the 4-week duration.

**Single-cell RNA sequencing (scRNA-seq)**. A375 cells were cultured with or without 1-μM dabrafenib in full-growth medium for 72 h before preparation of a single-cell suspension according to the 10X Genomics sample preparation protocol, 'Single-cell suspensions from cultured cell lines for scRNA-seq'. The single GEM capture, lysis, library construction and sequencing were performed by the Microarray and Genomics Core at the University of Colorado Anschutz Medical Campus. The untreated and the treated samples were prepared using the same chemical reagents on the same day and the libraries were sequenced in one lane of a NovaSEQ6000 with a sequencing depth of 400,000 reads/cell.

**Chromatin-bound MCM2 flow cytometry assay**. Fixed-cell immunostaining of chromatin-loaded MCM2 and subsequent flow cytometric analyses were performed as previously described[47]. Briefly, following the indicated treatments, cell suspensions were fixed in 4% paraformaldehyde and washed before stepwise staining: EdU click reaction with Alexa Fluor 488 (room temperature for 30 min), mouse anti-MCM2 immunostaining (BD Biosciences #610700 at 1:200 for 1 h at 37 °C), goat anti-mouse Alexa Fluor 546 immunostaining (Thermo Fisher #A-11003 at 1:500 for 1 h at 37 °C) and Hoechst 33342 staining (1:10,000; overnight at 4 °C). Final cell suspensions along with staining controls were analyzed using a BD FACSCelesta flow cytometer equipped with a 405, 488 and 561 nm lasers. FCS files for each sample were captured using FACSDiva and transferred to FlowJo for analysis.

**Comet assay with EdU incorporation**. The neutral comet assay was performed according to manufacturer protocol (Trevigen #4250-050-K) with an adaptation to capture EdU staining. Briefly, after treating cells for 30 min with EdU, cell suspensions were harvested and suspended in LM agarose gels on comet slides and processed for single-cell electrophoresis and DNA precipitation. To stain, gel sites were immersed in EdU click reaction cocktail with Alexa Fluor 647 for 30 min at room temperature. Slides were washed and then stained with SYBR Gold (Thermo Fisher, #S11494) for 30 min at room temperature. Slides were washed and dried thoroughly for 45 min before applying glycerol mountant and coverslips. Comet images were taken on a Nikon Ti-E using a 10×0.45 NA objective. Corresponding TIFF files were processed using the ImageJ plugin OpenComet (http://www.cometbio.org/), and EdU$^+$ cells were manually scored for each cell ID generated.

Any doublet events and incorrect comet head segmentations were omitted from the analysis.

**Quantification of FISH puncta**. RNA-FISH image analysis for *FOSL1, ETS1, MYC, CDC42EP1, RAB32* and *LINC01133* was performed using Harmony high-content imaging and analysis software. First, the DAPI channel containing Hoechst DNA stain was used to identify cell nuclei in each image. Then the total protein dye CF 568 succinimidyl ester in the Cy3 channel was used to create a whole-cell mask for each cell. Cells on the border of the image were eliminated from analysis. The spot-detection function was then applied to the RNA-FISH signal in the Cy5 channel and each RNA puncta was detected as an individual spot. The mean nuclear pRb S807/811 intensity was calculated from the FITC channel. The number of RNA puncta per cell and mean intensity of phospho-Rb were then exported from Harmony software into Matlab and plotted as a violin plot of number of puncta per cell. For the high-abundance *CCNA2* and *CCNB1* mRNAs, we used a different approach in which we measured the mean mRNA intensity in a 4-pixel ring around the nucleus.

**Statistical tests and violin plot visualization**. Statistical tests were performed using GraphPad Prism. For statistical differences in single-cell immunofluorescence, FISH marker and comet measurements represented as violin plots, *p* values were calculated using an unpaired *t*-test with Welch's correction for unequal variances between sample groups of all individual cells. For the number of mitoses quantified in the 12-day movie, the *p* value was calculated using a non-parametric Mann–Whitney rank test. For bar plots representing phospho-Rb$^+$ cell percentages among replicates, under-licensed cell percentages among replicates, apoptotic cell percentages among replicates or fold change in cell count among Geminin-sorted replicates, *p* values were calculated using an unpaired *t*-test. Significance levels are reported as *p* values < 0.05 (*), 0.01 (**), 0.001 (***) and 0.0001 (****) with corresponding star notations. Throughout the manuscript, 'ns' denotes no statistical significance.

For violin plots used throughout the paper, open circles represent the median values, and thick bars above and below each median represent the interquartile ranges of the distribution. The full distributions are displayed by the full range of the violin shape, with the width along the violin corresponding with the value frequency.

**Image processing and cell tracking for time-lapse movies**. Cells were tracked using EllipTrack[29]. In brief, EllipTrack segments cells by fitting nuclear contours with ellipses. EllipTrack then utilizes a machine learning algorithm to predict cell behaviours and maps ellipses between frames by maximizing the probability of cell lineage trees. Next, signals from each colour channel are extracted in the cell nuclei and cytoplasmic rings. Cell tracks were manually verified such that only cells correctly tracked during the entire movie were kept for downstream analysis. CDK2 activity was read out as the cytoplasmic:nuclear ratio of the DHB signal, as previously described[26]. Cell count over time was determined by the number of nuclei in the field of view at each time point.

**Analysis of single-cell CDK2 traces**. A customized script was used to determine whether a cell was proliferative or quiescent at each time point. In brief, we first identified 'seed regions' for proliferation, which were defined as the sets of continuous time points with CDK2 activity >0.9. Then, for each 'seed region', we searched the start (restriction point) and the end (mitosis point) of proliferation by examining the time points before and after the seed region, respectively. The restriction point was defined as the closest time point before the seed region with a slope of CDK2 activity <0.01. The mitosis point was defined as the closest time point after the seed region with a locally maximal H2B intensity. All time points between the restriction point and the mitosis point were assigned as proliferative, and the remaining time points were assigned as quiescent. The threshold for seed regions dropped to 0.6 for the final frames of the traces in order to identify cells that re-entered cell cycle but had yet to reach high CDK2 activity before movie ended. Finally, because the 0.9 threshold is quite high (corresponding to the time of S-phase entry), the algorithm might identify a part of G1 phase as quiescence if a cell is rapidly cycling, as in movies of untreated cells. We therefore converted all quiescence periods shorter than 4 h to proliferative to minimize misclassification, while maintaining high accuracy in classifying drug-treated cells.

For heatmaps, escapees and non-escapees were plotted separately and cells within each category were sorted by the similarity of their CDK2 traces with hierarchical clustering. The plots were then combined. Apoptotic cells are not included in the heatmaps. For heatmaps sorted on cell-cycle phase, cells from all categories were aggregated and sorted by the time of their first mitosis.

**Calculation of proliferation probability**. We computed the proliferation probabilities based on whether the mRNAs of 51 cell-cycle genes (Supplementary Data File 1) were detected or not. Cell-cycle genes were selected from CycleBase 3.0 database[65] to include signature genes of all cell-cycle phases. To compute the probability of proliferation, denote the fraction of proliferative cells in the population as $p_0$, determined experimentally by immunofluorescence staining of pRb S807/711. The values for untreated and dabrafenib-treated conditions were 0.95

and 0.18, respectively. Denote the probability of detecting at least one mRNA copy of cell-cycle gene $i$ in a proliferative cell as $p_i$. $p_i$ was computed by dividing the fraction of cells with at least one mRNA copy detected by $p_0$, and the value was kept between 0.01 and 0.99. Furthermore, assume that the probability of detection in a quiescent cell is $\varepsilon = 0.01$ (same for all genes). Assuming that mRNAs of different genes were independently detected, we have

$$P(\mathbf{X}|P) = \prod_i P(X_i|P) = \prod_i \left[ \delta_{X_i,0}(1-p_i) + \delta_{X_i,1}p_i \right] \quad (1)$$

and

$$P(\mathbf{X}|Q) = \prod_i P(X_i|Q) = \prod_i \left[ \delta_{X_i,0}(1-\epsilon) + \delta_{X_i,1}\epsilon \right] \quad (2)$$

Here, P and Q stand for 'proliferative' and 'quiescent'; $X_i$ is an indicator variable for detecting at least one mRNA copy of cell-cycle marker gene $i$; $\mathbf{X}$ is the vector of indicator variables ($\mathbf{X} = [X_1, X_2, \ldots, X_{51}]$); and $\delta$ is the Kronecker delta function. Using the fraction of proliferative cells in the population ($p_0$) as the prior probability, and using Bayes' theorem, we computed the probability of a cell being proliferative by

$$P(P|\mathbf{X}) = \frac{P(P)P(\mathbf{X}|P)}{P(P)P(\mathbf{X}|P) + P(Q)P(\mathbf{X}|Q)} = \frac{p_0 P(\mathbf{X}|P)}{p_0 P(\mathbf{X}|P) + (1-p_0)P(\mathbf{X}|Q)} \quad (3)$$

**scRNA-seq analysis**. Gene-expression matrices from the untreated and treated conditions were computed by Cell Ranger (10X Genomics) and combined into a single dataset by 'cellranger aggr'. The deeper-sequenced condition was randomly down-sampled such that two conditions had the same population-level library size in the combined dataset. Quality control was then performed such that invalid genes and outlier cells were removed. Here, a gene was denoted as invalid if it was not detected in at least one cell in both conditions, or if its gene symbol corresponded to multiple Ensemble IDs. A cell was denoted as an outlier if its log-library size or its log-number of detected genes was at least 5 median absolute deviation (MAD) lower than the population median, or if its percentage of expression mapped to mitochondrial genes was at least 5 MAD greater than the population median.

Proliferation probabilities of cells were computed as described above. Cells were classified into five groups based on their probabilities: subgroup 1, $P(P|\mathbf{X}) < e^{-60}$; subgroup 2, $P(P|\mathbf{X}) \in [e^{-60}, e^{-40})$; subgroup 3, $P(P|\mathbf{X}) \in [e^{-40}, e^{-20})$, subgroup 4: $P(P|\mathbf{X}) \in [e^{-20}, 1)$; and subgroup 5, $P(P|\mathbf{X}) = 1$. Cells belonging to subgroup 5 were denoted escapees and untreated proliferative cells (UT-P) and cells belonging to subgroup 1 and 2 were denoted non-escapees and untreated quiescent cells (UT-Q).

Differential gene-expression analysis was performed in Seurat[66]. In brief, the gene-expression matrix was normalized (method: 'LogNormalize', scaling factor: 10000) and scaled. The top 2000 highly variable genes were then detected (method: 'vst') and used for dimension reduction (principle component analysis). The top 15 principle components were used for clustering (Louvain algorithm)[67] and computation of coordinates on the t-SNE plot. Finally, differentially expressed genes (DEGs) for escapees vs. non-escapees and for escapees vs. UT-P were computed (Wilcoxon Rank Sum test, logfc.threshold: 0.25, min.pct: 0.01, adjusted $p$ value < 0.05).

The list of 40 intersecting genes was computed by intersecting the list of upregulated DEGs for escapees vs. non-escapees and the list of upregulated DEGs for escapees vs. UT-P. ATF4 targets were detected with the iRegulon plugin[40] in Cytoscape[68]. The overlap of the 40 intersecting genes with the Hallmark gene set[43] was computed on the Molecular Signatures Database website[69] (MSigDB).

Genes upregulated in escapees were defined as the upregulated (avg_logFC > 0) DEGs for escapees vs. non-escapees, and the genes upregulated in non-escapees were defined as the downregulated (avg_logFC < 0) DEGs for escapees vs. non-escapees. The overlap with previously published gene signatures was computed with the R package 'GeneOverlap'[70].

**Prognostic analysis**. The clinical data (as of January 5th, 2016) and mRNA data (normalized RSEM values of Tier 3 RNASeqV2) were retrieved from the Skin Cutaneous Melanoma (SKCM) Project of The Cancer Genome Atlas (TCGA) (The Cancer Genome Atlas Network, 2015). Patients ($n = 459$) with either a sequenced primary solid tumour sample (type '01') or a metastatic tumour sample (type '06') were used for the analysis. For patients with multiple samples sequenced ($n = 2$), the expression levels of the primary samples were used. Two genes in the 40-gene list (*LINC01133* and *TMSB4X*) were removed from the analysis due to the absence of measurements in the TCGA datasets. For each of the remaining genes, Cox proportional hazards analysis (survival ~ age + gender + tumour stage + gene-expression level) was performed. Six genes (*ACTB, CDC42EP1, RAB32, SLC3A2, SLC7A5* and *TUBB2A*) had significantly positive hazard ratios ($p < 0.05$) while none had significantly negative values (Supplementary Data File 2). For each of the six genes, patients with top and bottom 25% expression levels were used to compute Kaplan–Meier plots. Five genes (*CDC42EP1, RAB32, SLC3A2, SLC7A5* and *TUBB2A*) had a log-rank $p$ value below 0.05. In addition, for each of the five genes, the median survival of patients with top 25% expression levels was shorter than the median survival of patients with bottom 25% expression levels. This analysis

indicated that five genes in the 40-gene list correlated negatively with patient survival while none correlated positively. The Kaplan–Meier plots for the five genes are included in Fig. 6g, and the plots for the other genes are included in Supplementary Fig. 6.

To evaluate the enrichment of genes that had negative correlation on patient survival in our 40-gene list, we performed Cox proportional hazards analysis and computed Kaplan–Meier plots for all genes in the genome as described in the paragraph above. We found that 5.6% genes in the genome correlated negatively with patient survival (significantly positive hazard ratios; significant log-rank $p$ value for Kaplan–Meier plots; and patients with top 25% expression levels had shorter median survival than patients with bottom 25% expression levels). Meanwhile, 7.5% genes in the genome correlated positively (significantly negative hazard ratios; no requirement on the Kaplan–Meier plots or survival). Therefore, the probability of obtaining a better enrichment (at least five genes having negative correlation while none having positive correlation) through random sampling would equal to $\sum_{i=5}^{38} C_{38}^i 0.056^i (1 - 0.056 - 0.075)^{38-i} = 0.004$. Note that the two genes not found in the TCGA datasets were excluded from the calculation. This analysis showed that genes with long-term negative correlation on patient survival were indeed significantly enriched in our list of 40 genes upregulated in escapees.

**Reporting summary**. Further information on research design is available in the Nature Research Reporting Summary linked to this article.

## Data availability
Raw and processed scRNA-seq datasets are deposited in Gene Expression Omnibus (Accession Number: GSE164614). Raw and processed data are also available upon request. Source data are provided with this paper.

## Code availability
Scripts for tracking live-cell movies (EllipTrack)[29] are available at the GitHub repository https://github.com/tianchengzhe/EllipTrack. Scripts, input data and output data for scRNA-seq data analysis are deposited to the GitHub repository https://github.com/tianchengzhe/drug_response_scripts. Scripts for immunofluorescence and FISH quantification are available upon request.

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

## Acknowledgements

We thank Xuedong Liu, Kasey Couts, Grace Zheng and Rebecca Schweppe for comments on the manuscript; members of Spencer lab for general help and discussion; Theresa Nahreini and the cell culture facility for cell sorting; Joseph Dragavon at the BioFrontiers advanced light microscopy core; Kasey Couts and William Robinson at the University of Colorado Skin Cancer Biorepository. The results of Kaplan–Meier plots are in whole or part based upon data generated by the TCGA research network: https://www.cancer.gov/tcga. This work was conducted with the help and resources of the BioFrontiers

Computing Core at the University of Colorado BioFrontiers Institute. The PerkinElmer Opera Phenix is supported by NIH Grant 1S10ODO25072. The Aria Fusion FACS sorter and BD FACSCelesta are supported by NIH Grant S10ODO21601. This work was supported by a Kimmel Scholar Award (SKF16-126), a Searle Scholar Award (SSP-2016-1533) and a Beckman Young Investigator Award to S.L.S.

## Author contributions

C.Y. conducted the majority of experiments, analyses, data interpretation and manuscript preparation; C.T. developed EllipTrack, helped analyse microscopy and scRNA-seq data and assisted with manuscript preparation; T.E.H. assisted with experiments, analyses and manuscript preparation. C.Y., C.T., T.E.H. and N.K.J. manually verified cell traces; S.L.S. conceived the project, suggested the experiments, interpreted the data and wrote the manuscript with C.Y.

## Competing interests

The authors declare no competing interests.
