## [Peer Review File · Nature Communications]

REVIEWER COMMENTS

Reviewer #1 (Remarks to the Author):

In this paper 'Melanoma subpopulations that rapidly escape MAPK pathway inhibition rely on stress signaling and incur DNA damage', Yang et al used imaging-based approaches and single cell sequencing to study heterogeneous responses of melanoma cells to drug treatments and its underlying mechanisms. Understanding non-genetic reasons for drug resistance is essential for achieving durable therapeutic effects. The involvement of ATF4-mediated general stress response in drug tolerance demonstrated in this work is a novel discovery. Overall, this is a well written manuscript with high quality data. There are several questions to be addressed before it can be considered for publication.

Major points

- Prognostic analysis: it is problematic to combine primary and metastatic tumor samples together without considering influence of disease stage on survival. The rigorous method for prognosis is to use Cox proportional hazards regression model with adjustment for covariates (i.e. multi-variate Cox model). If significance is identified, KM plot is used to visualize the survival curves, and log-rank test is used to compare among groups. Also, what are the results of the other 32/40 genes? Are there genes beneficial to prognosis?
- Escapees out-proliferate non-escapees: the evidence supporting this conclusion is not sound. Fig 4e, what is the #mitoses comparison result in escapee (instead of KC+escapee) vs non-escapee? Also, seems mitoses of escapees mainly happen closer to 96hr time point, it is not clear whether escapees will continue proliferating and out-compete non-escapees given a longer observation time.
- What are the mechanisms for KC population? Those are potentially the ones most likely to contribute to resistance.

Minor points

- T-SNE plot needs clearer labeling (e.g. for T, UT) or description in legend.
- ATF4 is reported to have dual effects: pro-survival, and pro-death under persistent stress. The discovery of this work is consistent with prior reports, for drug tolerance as a short-/medium-term event. Suggest to add discussion for this part.
- The expression that '...genes...have a negative impact on patient survival' in the discussion is not correct. Prognostic analysis evaluates correlation between gene expression and survival, but does not indicate causal relationship.

Reviewer #2 (Remarks to the Author):

The manuscript by Chen Yang et al. provides novel and very interesting insight into the capacity of melanoma cells to evade MAPK pathway inhibitor cytotoxic effects.

Through very elegant, well designed experiments the authors give a detailed mechanism by which escapees cells continue proliferating under BRAF inhibition by acquiring an AXL^{high}/MITF^{low} transcriptional program mediated by mTORC1 activation.

Point to be addressed:

- While the data showing that the escapee subpopulation acquires AXL^{high}/MITF^{low} transcriptional program is very convincing it is not so clear why this population still shows an elevated MITF expression (almost 2 fold increase for the escapee population v approx. 2.5 fold for the non-escapee). Could the escapee population represent a heterogeneous population where MITF^{high} and low cells co-exist? This question needs to be addressed.
- Immunofluorescence experiments should easily determine if CDK2 positive cells correlate specifically with high AXL expressing cells and/or whether there exists a subpopulation of escapee cells where CDK2 activity correlates with its expression regulator MITF.
- What is the MITF expression in escapee cells surviving mTORC1 inhibition?
- Could this mixed-population be recapitulating the drug tolerance state from which "cooperative-resistance" has been postulated to arise? This needs to be addressed and thoroughly discussed.
- From the scRNAseq data: what is the "endothelin receptor status" of the escapee subpopulation?

Can they be separated between EDNRA+VE and EDNRB+VE cells? What happens when treating the escapee subpopulation with endothelin (A and/or B) receptor inhibitors?

- It would be necessary to determine if the escapee subpopulation can give rise to recurrent tumors. Given their dedifferentiated phenotype one could argue their capacity to regrow a tumor is significantly higher than that of drug-naïve cells (one would expect the non-escapee subpopulation to be incapable of growing tumors). An in vivo experiment with limiting dilutions of escapee cells v drug-naïve cells would shed light into this hypothesis. Do tumors originating from escapees cells recapitulate tumor heterogeneity = AXLhigh/MITFlow v AXLLow/MITFhigh ?

Reviewer #3 (Remarks to the Author):

In this manuscript, Yang et al use a combination of single-cell imaging, RNA-sequencing, bioinformatics analysis, and follow-up experiments to analyze the behavior of BRAF-mutant melanoma cells in response to 4 days of treatment with BRAF inhibitor. Their study (performed primarily in two melanoma cell lines A375 and WM278) identifies a subpopulation of cells that escape the effect of drug within three days. Non-genetic drug resistance in these cells is associated with the activation of mTORC1 and ATF4, incomplete licensing of replication origins, and elevated DNA damage. By correlating patient survival curves (from TCGA) to the expression of genes upregulated in A375 escapee cells, the authors show that these genes have a negative long-term impact on patient survival.

The overall topic of the manuscript and the single-cell methods utilized to investigate the problem of drug resistance/tolerance in melanoma cells are interesting in nature. However, I find it very difficult to identify how this manuscript would enhance our knowledge of drug resistance and fractional response to cancer therapy relative to what has already been described in literature. A major limitation is that the authors have founded their experiments and analyses based on a premise that ignores ~10 years of research on mechanisms of adaptive resistance and subpopulation response to BRAF/MEK inhibition. The outcome of the study is, therefore, a combination of findings that are not well-justified or connected and just replicate findings from other publications.

1) The emergence of subpopulations of slow-cycling, drug-insensitive cells within the first few days of BRAF inhibitor treatment and their reversible characteristics have been described by others who have used a variety of tools, including expression analysis of cell cycle genes/proteins and time-lapse imaging of cell cycle progression in individual cells (PMID: 25417704, 28069687).

2) Mechanisms of adaptive resistance to BRAF/MEK inhibition are diverse and vary across melanoma subtypes and thus may not be uncovered appropriately by studies that use only a couple of cell lines. For example, mTORC1 activity has been previously shown to be maintained after treatment with BRAF/MEK inhibitors (PMID: 23903755), especially in cell lines that have PTEN deletion such as WM278 (used in this study). Furthermore, among the most highly studied resistance mechanisms is the reactivation of the MAPK signaling pathway. MAPK reactivation may result either from the paradoxical activation of the pathway when monomer-selective BRAF inhibitors (such as vemurafenib or dabrafenib) are used, or as a consequence of relief of ERK-dependent feedbacks which lead to transcriptional/biochemical rewiring of the pro-growth signal transduction (PMID: 30770389, 23153539). Previous findings in these areas (which have not been considered in this manuscript) have led to the use of BRAF inhibitors in combination with MEK inhibitors (rather than BRAF inhibitors alone) in treatment of melanoma patients for a number of years now. Importantly, almost all of the previous studies have used A375 cells (which have been used as the primary melanoma model in this manuscript) to investigate such adaptive resistance phenomena. Thus, by focusing their analysis almost entirely on dabrafenib-treated A375 cells (e.g. Fig. 2), the authors have missed the opportunity to assess the contribution of MAPK pathway-related or MAPK-independent mechanisms toward the emergence of drug-tolerant cells.

3) In agreement with previous findings and reports, the authors show in Fig. 1 that the combination of dabrafenib with trametinib leads to a huge reduction in the percentage of drug-tolerant cells from ~40% (when A375 cells are treated with 1 uM dabrafenib alone) to only 3%

(when cells are exposed to 1 μ M dabrafenib + 10 nM trametinib). The authors use this small fraction of residual cells as a reason to argue that other factors should be responsible for the emergence of escapee cells and move on to identify such factors by single-cell RNA sequencing and find a potential role for mTORC1 and ATF4. However, they only test dabrafenib-treated cells (instead of dabrafenib plus trametinib-treated cells) by RNA sequencing. Furthermore, the combination of mTOR inhibitor or ATF4 knockdown with dabrafenib reduces the fraction of escapee (pRb-high) cells to ~10%, which is substantially higher than 3% as observed for dabrafenib plus trametinib. Taken together, it seems very difficult to identify the originality of the work or connect observations and the speculations/hypotheses that the authors present to justify the follow-up experiments.

4) Another major conclusion of this manuscript is the elucidation of the heightened DNA replication stress and DNA damage in drug-tolerant cells, suggesting that "a mutagenesis-prone, expanding subpopulation of cells may represent a reservoir for the development of permanent drug resistance". Previous studies (including, for example, PMID: 30709805) have investigated this issue in detail and show that MAPK pathway suppression may unmask latent DNA repair defects not only in BRAF-mutant melanomas, but also in NRAS-mutant and NF1-mutant cells. The mechanisms of this phenomenon have also been investigated across a variety of melanoma samples, including A375 cells and relevant mouse models. In addition, the authors' speculation/hypothesis that such an elevated DNA damage in drug escapes (but not in quiescent cells) may be responsible for the emergence of permanent drug resistance has not been tested in their study.

5) Melanoma tumors are known to harbor a diversity of genetic mutations as well as non-genetic subtypes as classified in TCGA studies (PMID: 26091043). A375 cells, for example, harbor CDKN2A mutations, which are likely to influence their cell cycle progression. PTEN is deleted in WM278 cells, and therefore may affect their Akt/mTOR response. Based on such diversity, studies of adaptive mechanisms that use only a couple of cell lines may not realistically capture the spectrum and relevance of these mechanisms and how they may or may not relate to pre-existing genetic mutations or epigenetic subtypes. In line with this comment, the authors do not observe significant cell death in response to 1 μ M dabrafenib treatment in their analysis of A375 and WM278 cells. They associate this finding with the cytostatic nature of this drug, which is not true. There is extensive evidence that BRAF inhibitors such as dabrafenib may induce substantial apoptosis in some BRAF-mutant cell lines but not in others.

6) The authors refer to TCGA data analysis to show that the expression of 8 out of the 40 ATF4/mTORC1-associated genes upregulated in A375 escapee cells are associated with poor patient survival. Although the effort toward clinical relevance is appreciated, this analysis has important limitations. First, it is not clear why the analysis does not cover the rest of 32 ATF4/mTORC1-associated genes? Second, the TCGA data analysis is based on pre-existing expression levels across patient tumors prior to MAPK targeted therapies. How does the analysis relate to the expression of these genes that are upregulated in A375 cells in response to dabrafenib treatment? The authors clearly conclude that they "failed to identify pre-existing cell states associated with escapees", and since escapees could not be eliminated by dabrafenib/trametinib treatment, they moved on to investigate adaptive mechanisms in A375 cells. The TCGA data analysis is, however, performed based on the pre-existing condition in melanoma tumors rather than investigating the drug-induced mechanisms.

Replies to Reviewers' Comments:

We thank the reviewers for their comments, which have ultimately made our findings more robust.

Reviewer #1 (Remarks to the Author):

In this paper 'Melanoma subpopulations that rapidly escape MAPK pathway inhibition rely on stress signaling and incur DNA damage', Yang et al used imaging-based approaches and single cell sequencing to study heterogeneous responses of melanoma cells to drug treatments and its underlying mechanisms. Understanding non-genetic reasons for drug resistance is essential for achieving durable therapeutic effects. The involvement of ATF4-mediated general stress response in drug tolerance demonstrated in this work is a novel discovery. Overall, this is a well written manuscript with high quality data. There are several questions to be addressed before it can be considered for publication.

Major points

- Prognostic analysis: it is problematic to combine primary and metastatic tumor samples together without considering influence of disease stage on survival. The rigorous method for prognosis is to use Cox proportional hazards regression model with adjustment for covariates (i.e. multi-variate Cox model). If significance is identified, KM plot is used to visualize the survival curves, and log-rank test is used to compare among groups. Also, what are the results of the other 32/40 genes? Are there genes beneficial to prognosis?

We thank the reviewer for the suggestion and agree that it is problematic to combine primary and metastatic tumor samples. We now included a more rigorous prognostic analysis. For the two patients with multiple tumor samples sequenced, we only used their primary tumor samples for the analysis. We applied Cox proportional hazards regression model (survival \sim age + gender + tumor stage + expression level) to the 40 uniquely upregulated genes. We identified six significant genes ($p < 0.05$), all of which had positive hazard ratios (negative for patient survival). We then computed the KM plots for these six genes and found that all but one had a log-rank p-value below 0.05. We therefore now claim that five genes, rather than eight genes in the original submission, have significant negative associations. In the revised manuscript, we include the KM plots for the five significant genes in Fig. 5g and include the plots for all other genes in Fig. S6. The hazard ratios are included in Table S2. The procedure of the analysis is also described in the "Prognostic Analysis" section of Methods.

In the original submission, we computed the KM plots for all 40 genes, but 32 of the 40 genes did not reach statistical significance (log-rank $p=0.05$) and we therefore did not include them in the manuscript. In our new analysis, we claim that the 33 insignificant genes (excluding the two genes not found in the TCGA datasets) should not be beneficial to prognosis.

- Escapees out-proliferate non-escapees: the evidence supporting this conclusion is not sound. Fig 4e, what is the #mitoses comparison result in escapee (instead of KC+escapee) vs non-escapee? Also, seems mitoses of escapees mainly happen closer to 96hr time point, it is not clear whether escapees will continue proliferating and out-compete non-escapees given a longer observation time.

We have now compared the mitosis events only between escapees vs. non-escapees as originally defined for the 10-day movie in Fig 4e (the original version of the manuscript), and we reached the same conclusion that escapees have a significantly higher number of mitoses than non-escapees.

of mitoses in each group are listed below:

Escapee: 1.9 ± 0.08 (mean \pm SEM); Median = 2

Non-escapee: 0.76 ± 0.06 (mean \pm SEM); Median = 0.5

We also made a new 8-day movie to see whether there is an increase in mitoses around the 96-hr time point, and indeed this was the case again. The CDK2 heatmap of the 8-day movie is shown here as Fig. R1.

With improvements to EllipTrack, we have now been able to track the movie from Fig. 4e for an additional two days, out to 12 days. The 12-day version of the experiment is now in Fig. 6e of the revised paper. We compared the # of mitoses in escapees vs. non-escapees between 120-288 hr (excluding the 96-120 hr timeframe where many mitoses were occurring), and we still observed significantly higher number of mitoses in escapees compared to non-escapees. This result has also been included in revised Fig. 6e.

We also conducted an additional experiment to corroborate the finding that escapees out-proliferate non-escapees over extended treatment: We sorted escapees and non-escapees into separate populations based on the Geminin expression after 3-day dabrafenib treatment, and continually cultured them in drug without drug holiday. By imaging these separate wells every few days, we found that Geminin⁺ cells grew better than Geminin⁻ cells over the one-month observation period. These data are presented in our revised Fig. 6f and discussed in the main text.

- What are the mechanisms for KC population? Those are potentially the ones most likely to contribute to resistance.

We thank the reviewer for pointing out this interesting observation. In the original manuscript, we did not distinguish KC and escapee cells in any fixed-cell measures since they cannot be distinguished by any means other than time-lapse microscopy. The original definition of a KC, escapee, or non-escapee cell was purely based on the time spent in quiescence while in drug before cell-cycle re-entry. This information can only be accessed by live-cell imaging data, but not in any fixed-cell assays.

Therefore, the immunofluorescence and scRNA-seq results in the paper are the representation of both KC and escapee populations together. In addition, the percentage of the KC population changed (or was eliminated) with different doses of drug treatment, indicating that the KC population is not a special subset of cells with a different type of resistance to treatment. Based on this, in the revised manuscript, we have unified the definition of escapees so that it is the same for fixed-cell and live-cell data types. Now, the KC cells are included in the escapee population, and the KC population is no longer singled out as a distinct population. The definition has been updated in the Methods to have no strict requirement

Fig. R1: Heatmaps of single-cell CDK2 traces in the 8-day movie. A375 cells were treated with 1 μ M dabrafenib at the beginning of the movie and continuously filmed for 8 days. Drug was refreshed every 2 days.

for the amount of time spent in quiescence before cell-cycle re-entry, and instead includes any cell that spends even a brief period in quiescence before cell-cycle reentry.

This new definition is clearly stated in the methods and is also listed here:

“Definition of escapees

In time-lapse microscopy experiments, escapees are defined as cells that re-enter the cell cycle after spending any amount of time in a drug-induced CDK2^{low} quiescence before re-entering the cell cycle. Non-escapees are defined as cells that enter into and remain in a drug-induced CDK2^{low} state until the end of the imaging period.”

Minor points

- T-SNE plot needs clearer labeling (e.g. for T, UT) or description in legend.

We agree with the reviewer that the t-SNE plot and population visualization was more complicated than it needed to be. We have updated the t-SNE with clearer labeling in the revised Fig. 3a.

- ATF4 is reported to have dual effects: pro-survival, and pro-death under persistent stress. The discovery of this work is consistent with prior reports, for drug tolerance as a short-/medium-term event. Suggest to add discussion for this part.

We agree with the reviewer and believe that ATF4 is a pro-survival pathway during early drug tolerance. We have added a short discussion of this point as follows:

“Our results show that cells can readily activate bypass pathways to re-enter the cell cycle. Indeed, activation of the ATF4 pathway represents an evolutionarily-conserved general stress response that may function in adaptation to other clinical MAPK inhibitors not examined here. ATF4 stress signaling is reported to have dual effects under persistent stress: pro-survival^{47,48} and pro-apoptotic^{49–54}. Here, we report the involvement of the ATF4 pathway in promoting drug escape and demonstrate the therapeutic potential of targeting this pathway by showing that ATF4 depletion reduces the escapee subpopulation, consistent with a pro-survival role. Thus, our characterization of the escapee phenotype and reliance on the ATF4-mediated stress response may have broad applicability, but much additional work is needed to determine how widespread these phenomena are.”

- The expression that ‘...genes...have a negative impact on patient survival’ in the discussion is not correct. Prognostic analysis evaluates correlation between gene expression and survival, but does not indicate causal relationship.

We agree with the reviewer that our original language incorrectly indicated a causal relationship. We have corrected the sentence to only reflect the correlational relationship between gene expression and survival.

This sentence has been updated in the text and also listed below:

“Second, interrogation of TCGA data showed that a strikingly high percentage of genes (5 of 40) upregulated in A375 escapees at 72 hr correlates negatively with patient survival assessed over decades while none correlated positively.”

Reviewer #2 (Remarks to the Author):

The manuscript by Chen Yang et al. provides novel and very interesting insight into the capacity of melanoma cells to evade MAPK pathway inhibitor cytotoxic effects.

Through very elegant, well designed experiments the authors give a detailed mechanism by which escapees cells continue proliferating under BRAF inhibition by acquiring an AXL^{high}/MITF^{low} transcriptional program mediated by mTORC1 activation.

Point to be addressed:

- While the data showing that the escapee subpopulation acquires AXL^{high}/MITF^{low} transcriptional program is very convincing it is not so clear why this population still shows an elevated MITF expression (almost 2 fold increase for the escapee population v approx. 2.5 fold for the non-escapee). Could the escapee population represent a heterogeneous population where MITF^{high} and low cells co-exist? This question needs to be addressed.

The MITF mRNA expression on average is almost 2-fold higher in escapees compared to untreated cells. However, the MITF protein expression only has a 1.03-fold difference between escapees and non-escapees. We agree with the reviewer that the escapee population is a heterogeneous population where MITF-high and MITF-low cells co-exist. Our evidence for this includes:

1. In the t-SNE plot in Fig. S4b, both MITF-high and MITF-low cells exist in the escapee peninsula.
2. In the immunofluorescence data in Fig. S4c, the orange violin representing escapees has mostly lowish MITF but there is a long tail with some cells having high MITF, indicating heterogeneous expression of MITF in escapees.

- Immunofluorescence experiments should easily determine is CDK2 positive cells correlate specifically with high AXL expressing cells and/or whether there exists a subpopulation of escapee cells where CDK2 activity correlates with its expression regulator MITF.

Yes, in Fig. S4c, the pRb⁺ (CDK2 positive) cells in the orange violin correlate with high AXL expression and lower MITF expression. Note that there is a long tail of both genes' expression, indicating the heterogeneous nature of the escapee population.

- What is the MITF expression in escapee cells surviving mTORC1 inhibition?

The escapees surviving mTORC1 inhibition still show lower expression of MITF than non-escapees, indicating that the escapees surviving both treatments may share similar MITF^{low} gene signature. The split violin plot for this experiment is shown on the right.

- Could this mixed-population be recapitulating the drug tolerance state from which “cooperative-resistance” has been postulated to arise? This needs to be addressed and thoroughly discussed.

We thank the reviewer for bringing up the interesting concept of cooperative resistance. We believe the question can be rephrased as: can an escapee help a non-escapee develop drug tolerance and cell-cycle re-entry? We examined our data for this possibility by analyzing the spatial distribution of escapees, but

we did not see evidence for a cooperative effect from escapee population under our experimental setup. One possible reason could be that this time scale is too short for cooperative resistance to develop. In this study, we only focused on the initial 4-day treatment period, which is relatively short compared to other studies on cooperative resistance, and thus we do not rule out the possibility of escapees influencing nearby cells during longer-term treatment or in *in vivo* settings.

We have updated the Discussion with a section on this point:

“Cooperative resistance could also play a role in drug escape, wherein one cell can release stimulatory factors that promote the proliferation of nearby cells^{57,58}. However, analysis of spatial correlations in our imaging datasets found no evidence linking escapees to cell-cycle entry of neighboring cells, although we do not rule out the possibility of an effect during extended treatment or in *in vivo* settings.”

- From the scRNAseq data: what is the “endothelin receptor status” of the escapee subpopulation? Can they be separated between EDNRA+VE and EDNRB+VE cells? What happens when treating the escapee subpopulation with endothelin (A and/or B) receptor inhibitors?

We plotted EDNRA and EDNRB expression on the tSNE plot as shown below. EDNRA is barely expressed (both untreated and dabrafenib-treated cells). As for EDNRB, most escapees have low or no expression. Within the 85 escapees, only two escapees express EDNRA and 22 escapees express EDNRB. Thus, the escapee population cannot be separated by EDNRA and EDNRB expression.

Since the majority of the cells do not express EDNRA, we only treated cells with EDNRB inhibitor BQ788 combined with dabrafenib to see whether further suppression of EDNRB would suppress the escapee population. The percentage of escapee population is not different in dabrafenib alone or dabrafenib combined with BQ788, indicating that EDNRB expression does not contribute to the escape phenotype (please see the bar plot on the right). This is consistent with our scRNA-seq data.

- It would be necessary to determine if the escapee subpopulation can give rise to recurrent tumors. Given their dedifferentiated phenotype one could argue their capacity to regrow a tumor is significantly higher than that of drug-naïve cells (one would expect the non-escapee subpopulation to be incapable of growing tumors). An *in vivo* experiment with limiting dilutions of escapee cells v drug-naïve cells would shed light into this hypothesis. Do tumors originating from escapee cells recapitulate tumor heterogeneity = AXLhigh/MITFlow v AXLow/MITFhigh?

We agree that such an *in vivo* study would be nice to have. However, there are a number of technical challenges that make it too difficult to complete this experiment:

- a. Sorting escapees based on a snapshot in time will exclude cells that are going to escape later and escapees that just finished their cell cycle. Thus, we would not be able to compare a pure population of escapees and non-escapees (non-escapees would include some would-be escapees either before or after the point of observation). In order to catch all the escapees, we would ideally collect all cells that re-entered the cell cycle from days 2-7 after drug treatment; however, there is no current sensor or tool to record the cell-cycle history of a population. While we have done this type of sorting for our colony outgrowth experiment (Fig. 6f), a pure population of non-escapees would be more critical in an *in vivo* experiment where these small differences would have a substantial impact on the system. This pure population comparison is especially important when working with animal models that are costly and subject to ethical control and lower sample sizes.
- b. Any time spent out of drug after sorting allows cells to revert to the parental state. So the escapees and non-escapees would have to be sorted directly into drug and immediately implanted in a dabrafenib-treated mouse. However, a significant number of cells die if they are sorted directly into drug from the combined stress of sorting and drug treatment. Therefore, the number of cells implanted into a mouse would be much harder to control than in our *in vitro* collagen-coated monolayer system, making the cells accepted and retained by the animal much more variable, and quite possibly making the results unreliable. In addition, whether pre-treatment of dabrafenib in a mouse would prime the mouse for drug resistance for further treatment is unknown.
- c. Due to the COVID-19 pandemic, it will be difficult to set up a new collaboration with a lab at the medical school to do *in-vivo* experiment since all University of Colorado labs are at 50% capacity and barely able to do their own core experiments, and will likely remain this way for the foreseeable future.

Reviewer #3 (Remarks to the Author):

In this manuscript, Yang et al use a combination of single-cell imaging, RNA-sequencing, bioinformatics analysis, and follow-up experiments to analyze the behavior of BRAF-mutant melanoma cells in response to 4 days of treatment with BRAF inhibitor. Their study (performed primarily in two melanoma cell lines A375 and WM278) identifies a subpopulation of cells that escape the effect of drug within three days. Non-genetic drug resistance in these cells is associated with the activation of mTORC1 and ATF4, incomplete licensing of replication origins, and elevated DNA damage. By correlating patient survival curves (from TCGA) to the expression of genes upregulated in A375 escapee cells, the authors show that these genes have a negative long-term impact on patient survival.

The overall topic of the manuscript and the single-cell methods utilized to investigate the problem of drug resistance/tolerance in melanoma cells are interesting in nature. However, I find it very difficult to identify how this manuscript would enhance our knowledge of drug resistance and fractional response to cancer therapy relative to what has already been described in literature. A major limitation is that the authors have founded their experiments and analyses based on a premise that ignores ~10 years of research on mechanisms of adaptive resistance and subpopulation response to BRAF/MEK inhibition. The outcome of the study is, therefore, a combination of findings that are not well-justified or connected and just replicate findings from other publications.

1) The emergence of subpopulations of slow-cycling, drug-insensitive cells within the first few days of BRAF inhibitor treatment and their reversible characteristics have been described by others who have used a variety of tools, including expression analysis of cell cycle genes/proteins and time-lapse imaging of cell cycle progression in individual cells (PMID: **25417704**, **28069687**).

While there are prior reports on the topic of slow-cycling, drug-insensitive cells within a few days of BRAF inhibitor treatment, there are still many questions left unanswered and the topic remains under-appreciated in the field. We cited multiple papers in the original manuscript about drug tolerance, including Sharma et al., 2010, Roesch et al., 2010, Fallahi-Sichani et al., 2017 (PMID: **28069687**, suggested by reviewer, but already cited in original submission), and Smith et al., 2016. In the revised manuscript, we added Ravindran et al., 2015 (PMID: **25417704**) as the reviewer suggested, as this is a very good paper as well.

The novelty of our work compared to the existing work (including PMID: 25417704, 28069687) lies in two aspects:

1. We studied drug tolerance at the single-cell level using live-cell imaging and single-cell RNA sequencing.

Compared to previous research, where drug tolerance was mainly studied using bulk populations before and after treatment, we utilized single-cell time-lapse microscopy to monitor longitudinal single-cell responses to drug treatment over time. This enabled us to visualize extensive heterogeneity in cancer-cell drug responses, which cannot be seen by bulk population studies, nor be fully appreciated by scRNA-seq.

2. We focused on the *proliferation vs. quiescence* decision among the drug-treated survivors in drug, instead of comparing untreated cells with drug-treated cells in previous research.

The majority of existing literature is essentially focused on comparing untreated parental cells to cells not killed by the drug treatment, often referred to as “drug-tolerant persister” cells. However, these studies do not examine heterogeneity within the un-killed drug-tolerant population.

One important criterion for drug resistance is whether cells can proliferate in treatment or not. We and many others postulate that proliferation is critically important for the drug adaptation process since proliferation can expand the population and potentially enable cells to acquire mutations via faulty replication during S phase. In this study, we are specifically looking for the mechanisms that enable proliferation in drug by comparing escapees and non-escapees within the drug-treated condition. We also compare escapees to their untreated proliferative counterparts. Thus, our findings are uniquely targeted to *heterogeneity in proliferation potential in drug*. Our findings that escapees incur increased DNA damage and yet out-proliferate non-escapees over extended treatment add a new layer to the theory of drug tolerance and may very well define an elusive root of permanent drug resistance.

2) Mechanisms of adaptive resistance to BRAF/MEK inhibition are diverse and vary across melanoma subtypes and thus may not be uncovered appropriately by studies that use only a couple of cell lines. For example, mTORC1 activity has been previously shown to be maintained after treatment with BRAF/MEK inhibitors (PMID: 23903755), especially in cell lines that have PTEN deletion such as WM278 (used in this study). Furthermore, among the most highly studied resistance mechanisms is the reactivation of the MAPK signaling pathway. MAPK reactivation may result either from the paradoxical activation of the pathway when monomer-selective BRAF inhibitors (such as vemurafenib or dabrafenib) are used, or as a consequence of relief of ERK-dependent feedbacks which lead to transcriptional/biochemical rewiring of the pro-growth signal transduction (PMID: 30770389, **23153539**). Previous findings in these areas (which have not been considered in this manuscript) have led to the use of BRAF inhibitors in combination with MEK inhibitors (rather than BRAF inhibitors alone) in treatment of melanoma patients for a number of years now. Importantly, almost all of the previous studies have used A375 cells (which have been used as the primary melanoma model in this manuscript) to investigate such adaptive resistance phenomena. Thus, by focusing their analysis almost entirely on dabrafenib-treated A375 cells (e.g. Fig. 2), the authors have missed the opportunity to assess the contribution of MAPK pathway-related or MAPK-independent mechanisms toward the emergence of drug-tolerant cells.

We thank the reviewer for pointing out this paper (PMID: 23903755) to us, where the effectiveness of combining a BRAF inhibitor and mTORC1 inhibitor in melanoma patients was reported. While we accept the lack of novelty in mTORC1 contributions in our drug tolerance system, mTORC1 was not the main focus of our study. We are, however, glad to further corroborate the findings of others and are encouraged to see reproducible results across experimental systems and lab environments. Aside from the results in A375 and WM278 shown previously, we also detected increased mTORC1 activity in an *ex vivo* culture of melanoma patient line MB3883 as shown in Fig. 5d in the current manuscript. Furthermore, we treated two additional melanoma cell lines in our revised manuscript, SKMEL28 and SKMEL19, with dabrafenib and rapamycin and observed a reduction in escapee population (see bar plot below). These results shown that the upregulation of mTORC1 pathway is not restricted to cell lines that with PTEN mutations, but rather could be more general than previously appreciated.

Besides involvement of the mTORC1 pathway, we also identified the upregulation of ATF4 stress signaling in escapees. ATF4 involvement in rapid escape, and stress signaling more broadly, is the more novel discovery that we focused on in our original submission. This conclusion is now further emphasized in our revised manuscript, where we see this phenomenon occur across additional cell lines.

We are familiar with the work on MAPK pathway reactivation as a resistance mechanism and we agree that this is a prominent occurrence across targeted therapies. We had already cited a number of papers on MAPK pathway reactivation in response to BRAF inhibition in the original manuscript: Roesch et al.; Fallahi-Sichani et al., 2017; Lito et al., 2012 (PMID: **23153539**, suggested by reviewer but already cited in original submission); Hirata et al., 2015; and Smith et al., 2016. But we agree that we could cite more papers and devote more text to this topic. In the revised manuscript, we have cited more key papers on MAPK pathway reactivation in driving drug tolerance. Please refer the first paragraph in revised manuscript and citations 10 -19.

We respectfully disagree with the reviewer that we missed the opportunity to assess the contribution of MAPK pathway-related or MAPK-independent mechanisms. We have internal data that agrees with existing literature on MAPK pathway reactivation following BRAF inhibition, but we specifically avoided getting into MAPK-related mechanisms, because as the reviewer points out, reactivation of the MAPK pathway is the most highly studied resistance mechanism to BRAF inhibition. Instead, we focused specifically on MAPK-independent mechanisms and identified ATF4 stress signaling as important for escape from drug-induced quiescence. While we discovered this result in A375 melanoma cells, we have now validated it in a total of four melanoma cell lines (WM164, WM278, SKMEL28, A375; Fig. 3d) and a patient biopsy (Fig. 5c). Increased ATF4 in escapees relative to non-escapees is also observed beyond just treatment with dabrafenib, with vemurafenib, PLX, 8394, and trametinib (Fig. S7a). Therefore, we have made a novel discovery about upregulation of ATF4 stress signaling (and stress responses more broadly based on our scRNA-seq data) in escapees across a variety of cell lines and targeted therapies. This discovery has been emphasized more in our revised manuscript.

3) In agreement with previous findings and reports, the authors show in Fig. 1 that the combination of dabrafenib with trametinib leads to a huge reduction in the percentage of drug-tolerant cells from ~40% (when A375 cells are treated with 1 uM dabrafenib alone) to only 3% (when cells are exposed to 1 uM dabrafenib + 10 nM trametinib). The authors use this small fraction of residual cells as a reason to argue that other factors should be responsible for the emergence of escapee cells and move on to identify such factors by single-cell RNA sequencing and find a potential role for mTORC1 and ATF4. However, they only test dabrafenib-treated cells (instead of dabrafenib plus trametinib-treated cells) by RNA sequencing. Furthermore, the combination of mTOR inhibitor or ATF4 knockdown with dabrafenib reduces the fraction of escapee (pRb-high) cells to ~10%, which is substantially higher than 3% as observed for dabrafenib plus trametinib. Taken together, it seems very difficult to identify the originality of the work or connect observations and the speculations/hypotheses that the

authors present to justify the follow-up experiments.

The originality of the paper is to find the mechanisms of early cell-cycle re-entry in the presence of dabrafenib. We indeed also see reactivation of MAPK signaling in response to dabrafenib, consistent with previous findings, but this topic has been extensively studied. Instead, we focused on MAPK-independent mechanisms that enable proliferation in dabrafenib using single-cell RNA sequencing technology.

There are technical reasons why we did not perform scRNA-seq on cells treated with dabrafenib + trametinib. To identify escapees based on their single-cell transcriptome profile, we used a gene list that contains 51 cell cycle genes to calculate the proliferation probability for each cell. This probability distribution was not bimodal for the population as it is in immunofluorescence experiments, due to:

1. The well-known drop-out issue associated with scRNA-seq.
2. Subtle difference of gene expression in different cell cycle states. Escape from dabrafenib-induced quiescence is a continuous process as cells need to upregulate different genes at different time points. For example, a cell that is going to escape from drug in 10 hr will have a very different transcriptome compared to a cell that is going to escape in 2 hr, even though both cells are in quiescence at the time of observation. Thus, when we use a fixed-cell assay, both cells will be detected as non-escapees, but the proliferation probability calculated by the transcriptome profile will be very different.

Due to the technical issues listed above, we only identified about 2% high-confidence escapees from scRNA-seq, even though we know the integrated number of escapees by the end of 96hr of time-lapse imaging is 40%. Our conservative definition of escapees, however, allows us to make high-confidence conclusions about their transcriptomic profile. Under dabrafenib plus trametinib condition, there are only 3% escapees by live-cell imaging. So, the percentage of escapees identified in the double-drug population by scRNA-seq would be about 0.15%, which is far too low to draw any conclusion from the data. Additionally, some patient subsets are still treated with BRAF inhibition alone to avoid the overt toxicity that MEK inhibitors can introduce, and these conditions of resistance are of direct relevance to our scRNA-seq dataset. Expanding on this, since we acknowledge that dabrafenib monotherapy is certainly not the only treatment condition used for these patients, we dedicated a wide variety of experiments across cell lines and MAPK pathway inhibitors, alone and in combination, to see if our identified stress signaling was a prominent occurrence in other types of escapees. Indeed, as mentioned before, we see ATF4-mediated signaling also in escape from high doses of vemurafenib, PLX8394, trametinib, and trametinib plus dabrafenib; this is true across several cell lines and in patient cultures.

We agree with the reviewer that dab+tra is more effective than dab+rapamycin or dab+siATF4. However, this does not mean that the mTORC1 and ATF4 pathways are not playing a role in the escapee phenomenon. As we show in Fig. 3e and Fig. S5a, co-treatment of dab+rapamycin or dab+siATF4 decrease the escapee population in multiple melanoma cell lines. To test whether escapees from dab+tra condition also upregulate the mTORC1 and ATF4 pathways, we treated cells with either a combination of dab+tra+rapamycin or dab+tra+siATF4, and we found a further reduction in escapees in the triple-drug conditions compared to mono- and double-drug conditions in two melanoma cell lines shown below. The difference between the significance level is due to the sample size difference. Together, these results suggest that cells escaping from the dab+tra condition also upregulate the mTORC1 and ATF4 pathways.

4) Another major conclusion of this manuscript is the elucidation of the heightened DNA replication stress and DNA damage in drug-tolerant cells, suggesting that "a mutagenesis-prone, expanding subpopulation of cells may represent a reservoir for the development of permanent drug resistance". Previous studies (including, for example, PMID: 30709805) have investigated this issue in detail and show that MAPK pathway suppression may unmask latent DNA repair defects not only in BRAF-mutant melanomas, but also in NRAS-mutant and NF1-mutant cells. The mechanisms of this phenomenon have also been investigated across a variety of melanoma samples, including A375 cells and relevant mouse models. In addition, the authors' speculation/hypothesis that such an elevated DNA damage in drug escapes (but not in quiescent cells) may be responsible for the emergence of permanent drug resistance has not been tested in their study.

We thank the reviewer for pointing out this paper (PMID: 30709805), which is a good paper that we were not aware of. We did cite a similar paper describing DNA damage upon MAPK inhibition showing direct results on genomic instability in the long term (Russo et al., 2019). While both papers show more evidence of the long-term impact of this damage than we do, they only focused on the DNA damage effects in bulk populations. Our study takes these results one step further by pinning down the subset of cells experiencing the damage – those that are attempting to cycle in drug.

The speculation that 'elevated damage in escapees may eventually lead to drug resistance' was only meant to be a point of discussion. We have now removed any even slightly speculative statements from the main text, and only include these as possible implications of the work in the Discussion. Fully proving the link between escapees and eventual bona fide drug resistance is well beyond the scope of this study, as testing this would require labeling the tumor cells with unique barcodes, identifying all escapees that appear over a 2-7 day period in drug, and then culturing them until genetic mutations rise. We agree that this point is important, and plan to test this in a future study.

5) Melanoma tumors are known to harbor a diversity of genetic mutations as well as non-genetic subtypes as classified in TCGA studies (PMID: 26091043). A375 cells, for example, harbor CDKN2A mutations, which are likely to influence their cell cycle progression. PTEN is deleted in WM278 cells, and therefore may affect their Akt/mTOR response. Based on such diversity, studies of adaptive mechanisms

that use only a couple of cell lines may not realistically capture the spectrum and relevance of these mechanisms and how they may or may not relate to pre-existing genetic mutations or epigenetic subtypes. In line with this comment, the authors do not observe significant cell death in response to 1 uM dabrafenib treatment in their analysis of A375 and WM278 cells. They associate this finding with the cytostatic nature of this drug, which is not true. There is extensive evidence that BRAF inhibitors such as dabrafenib may induce substantial apoptosis in some BRAF-mutant cell lines but not in others.

We agree with the reviewer that dabrafenib induces substantial apoptosis in some lines but more of a cytostatic effect in other lines, depending on genetic background. To address this point, we have now tested the dabrafenib response in six different melanoma cell lines with different genotypes:

SKMEL267C; WM164; WM278; SKMEL28; A375; SKMEL19

We observed a full range of drug response when monitoring apoptosis at three time points: 2, 4, and 14 days of dabrafenib treatment. For example, SKMEL267C is remarkably sensitive to BRAF inhibition, with 94% of cells dying after 96 hr treatment. On the other end of the spectrum, SKMEL19 is incredibly resistant to BRAF inhibition with only 8% death and little reduction in phospho-Rb status after 96 hr treatment. Please see Fig. S1c for full results.

We also updated the text to reflect this change in the paper:

“Apoptosis assays detected a large range of cell death across six cell lines, ranging from 94% in SKMEL267C to 17% in A375 to 8% in SKMEL19 cells after 96 hr of 1 μ M dabrafenib treatment, consistent with previous findings that BRAF inhibitors induce substantial apoptosis in some cell lines, but not in others⁶ (Supplementary Fig. 1b and c). Cell lines showing minimal apoptosis nevertheless still respond to the drug to varying degrees, visible as a reduction of cell proliferation.”

We also performed dose-response experiments in five different commercial melanoma cell lines and two *ex-vivo* culture of patient cells to further illustrate the point that different cell lines have different sensitivity ranges for dabrafenib. These data are shown in Fig. 1c and 5a.

6) The authors refer to TCGA data analysis to show that the expression of 8 out of the 40 ATF4/mTORC1-associated genes upregulated in A375 escapee cells are associated with poor patient survival. Although the effort toward clinical relevance is appreciated, this analysis has important limitations. First, it is not clear why the analysis does not cover the rest of 32 ATF4/mTORC1-associated genes? Second, the TCGA data analysis is based on pre-existing expression levels across patient tumors prior to MAPK targeted therapies. How does the analysis relate to the expression of these genes that are upregulated in A375 cells in response to dabrafenib treatment? The authors clearly conclude that they “failed to identify pre-existing cell states associated with escapees”, and since escapees could not be eliminated by dabrafenib/trametinib treatment, they moved on to investigate adaptive mechanisms in A375 cells. The TCGA data analysis is, however, performed based on the pre-existing condition in melanoma tumors rather than investigating the drug-induced mechanisms.

We thank the reviewer for this comment. For the first point, we computed the KM plots for all 40 genes, but 32 genes did not reach statistical significance (log-rank $p < 0.05$) and we therefore did not include their KM plots in the original submission. In the revised manuscript, we performed a more rigorous prognostic analysis (refer to our response to Reviewer 1, Point 1). We now include the KM plots for the five significant genes in Fig. 5g as well as the plots for all other genes in Fig. S6.

We agree with the reviewer's second point that it would be nice to analyze datasets in drug-treated patient samples that are linked to patient survival. However, these types of patient cohorts are very limited, and we only identified two that fit our criteria in the literature (Cohort 1: Nat Commun. 2014 Dec 02; 5:5694; Cohort 2: Clin Cancer Res. 2017 Oct 15; 23:6054-6061; both from Helen Rizos' lab). We applied Cox proportional hazards regression model (Survival ~ age + gender + expression level) to the 9 patients in Cohort 1 (one extra patient was reported in the Cohort but its dataset was not uploaded to GEO). Here, the overall survival and the expression levels in the progressed tumor samples were used (very few patients had early-treated tumor samples measured). By setting a threshold p-value of 0.1, we identified six significant genes (CDC42EP1, ACTB, UGCG, SLC3A2, HEBP2, and HMOX1), all of which had negative associations with patient survival. Importantly, three of these genes (CDC42EP1, ACTB, and SLC3A2) were also found significant in the TCGA datasets (Cox regression), and one of the genes we already highlight in the manuscript (CDC42EP1) was identified as well. Similar results were obtained when applying the analysis to Cohort 2. We therefore believe that our prognostic analysis is relevant to the study.

REVIEWER COMMENTS

Reviewer #1 (Remarks to the Author):

The authors answered all of my previously raised concerns, and clarified the definition of escapees. Also, the modified Fig 1 further highlights the fractional killing issue of cancer therapeutics, and better motivates the rest of the work in the manuscript. This is a beautiful work, and I recommend it for a publication in Nature Communications now.

Reviewer #2 (Remarks to the Author):

The authors have correctly clarified the points raised in my previous revision.

Reviewer #3 (Remarks to the Author):

The authors' revisions have improved the manuscript. The unique results of the study are illustrated more clearly with the additional figures and analyses throughout the manuscript. Nevertheless, two of my major concerns (points 2 and 6) have not been addressed adequately. Without addressing these points, there is a disconnection between the results and conclusions of this manuscript.

Related to Concern 2:

Reading the revised manuscript carefully and considering the authors' response to the comment, there is still inconsistency between the authors' data and their interpretation of whether MAPK-related or MAPK-independent mechanisms lead to the emergence of escapees. In their response to the comment, the authors mentioned that "We specifically avoided getting into MAPK-related mechanisms, because reactivation of the MAPK pathway is the most highly studied resistance mechanism to BRAF inhibition. Instead, we focused specifically on MAPK-independent mechanisms". The single-cell sequencing analysis, however, has been performed on cells treated with a BRAF inhibitor (dabrafenib at 1 μ M), a condition in which (based on extensive literature) cells are expected to have suboptimal MAPK inhibition due to the pathway rebound within the first few hours of treatment (see PMID: 23153539 and many more). How is it possible to extract "MAPK-independent" mechanisms of drug escape without either performing the analysis on fully MAPK-inhibited cells, or confirming that the escapees do not already have active MAPK signaling, or demonstrating that the identified pathways are not under the control of MAPK signaling in any way? While the authors insist on investigating MAPK-independent mechanisms, they show no measurement of MAPK signaling in the escapee cells throughout the manuscript. The only measurement of ERK phosphorylation levels is performed at an extremely high concentration (10 μ M shown in the supplements), which is much higher than the concentration used for single-cell sequencing and it is done by Western blotting that cannot distinguish escapees from non-escapees. This is also inconsistent with the authors' argument that the novelty of their work is in the application of single-cell tools. I believe this is a key point that should be resolved given that MAPK pathway inhibition is the core of the manuscript's argument, including its title. I acknowledge the difficulty of performing single-cell sequencing analysis on cells treated with the combination of dabrafenib and trametinib. However, the authors can at least check the expression of MAPK-related genes in the subpopulation of escapees shown in Fig. 3a to see if MAPK signaling is involved. Also, the authors argue that the "subpopulation of escapees" relies on stress signaling mediated by ATF4. This conclusion is based on the filters that they have set on the gene set enrichment analysis (Fig. 3b, c) to identify the genes that are differentially expressed in "both" dabrafenib-treated escapees vs. non-escapees and in dabrafenib-treated escapees vs. untreated proliferating cells. Based on this analysis, one would expect that untreated proliferating cells or dabrafenib-treated non-escapees should be insensitive to ATF4 knockdown. However, ATF4 siRNA data are only shown for dabrafenib-treated cells but not for untreated cells or for escapees vs. non-escapees (Fig. 3e).

Related to Concern 6:

The prognosis statistical analysis should be improved in order to support the authors' conclusion

about the significance of ATF4/mTORC1-associated genes for patient survival. The authors show that only 5 out of 40 genes correlate negatively with patient survival. The question is: how many genes out of any 40 randomly selected genes would correlate negatively with patient survival using the exact same approach? This is important to address, as authors show that 10% and 7.5% genes in the genome have significantly positive and negative hazard ratios. Therefore, for authors' conclusion (about the clinical relevance of ATF4/mTORC1-associated genes) to have statistical significance, a permutation analysis should be performed, based on which p-values should be corrected.

Response to Reviewer 3

Reviewer #1 (Remarks to the Author):

The authors answered all of my previously raised concerns, and clarified the definition of escapees. Also, the modified Fig 1 further highlights the fractional killing issue of cancer therapeutics, and better motivates the rest of the work in the manuscript. This is a beautiful work, and I recommend it for a publication in Nature Communications now.

Reviewer #2 (Remarks to the Author):

The authors have correctly clarified the points raised in my previous revision.

Reviewer #3 (Remarks to the Author):

The authors' revisions have improved the manuscript. The unique results of the study are illustrated more clearly with the additional figures and analyses throughout the manuscript. Nevertheless, two of my major concerns (points 2 and 6) have not been addressed adequately. Without addressing these points, there is a disconnection between the results and conclusions of this manuscript.

Related to Concern 2:

Reading the revised manuscript carefully and considering the authors' response to the comment, there is still inconsistency between the authors' data and their interpretation of whether MAPK-related or MAPK-independent mechanisms lead to the emergence of escapees. In their response to the comment, the authors mentioned that "We specifically avoided getting into MAPK-related mechanisms, because reactivation of the MAPK pathway is the most highly studied resistance mechanism to BRAF inhibition. Instead, we focused specifically on MAPK-independent mechanisms". The single-cell sequencing analysis, however, has been performed on cells treated with a BRAF inhibitor (dabrafenib at 1uM), a condition in which (based on extensive literature) cells are expected to have suboptimal MAPK inhibition due to the pathway rebound within the first few hours of treatment (see PMID: 23153539 and many more). How is it possible to extract "MAPK-independent" mechanisms of drug escape without either performing the analysis on fully MAPK-inhibited cells, or confirming that the escapees do not already have active MAPK signaling, or demonstrating that the identified pathways are not under the control of MAPK signaling in any way? While the authors insist on investigating MAPK-independent mechanisms, they show no measurement of MAPK signaling in the escapee cells throughout the manuscript. The only measurement of ERK phosphorylation levels is performed at an extremely high concentration (10 uM shown in the supplements), which is much higher than the concentration used for single-cell sequencing and it is done by Western blotting that cannot distinguish escapees from non-escapees. This is also inconsistent with the authors' argument that the novelty of their work is in the application of single-cell tools. I believe this is a key point that should be resolved given that MAPK pathway inhibition is the core of the manuscript's argument, including its title. I acknowledge the difficulty of performing single-cell sequencing analysis on cells treated with the combination of dabrafenib and trametinib. However, the authors can at least check the expression of MAPK-related genes in the subpopulation of escapees shown in Fig. 3a to see if MAPK signaling is involved.

We certainly agree with the literature that MAPK pathway reactivation contributes to the emergence of escapees in the presence of BRAF inhibition, and agree that the paper would benefit from demonstrating this point experimentally rather than simply citing the literature as we did previously. We now dedicated a new main figure (revised Fig. 3) to show that MAPK signaling indeed is reactivated in escapees relative to non-escapees,

consistent with previous findings from other groups. We used single-cell RNA-FISH to measure three MAPK pathway downstream targets, *Fos11*, *Ets1*, and *Myc* in dabrafenib-treated escapees and non-escapees and found significantly increase expression in escapees (revised Fig. 3a), indicating that MAPK pathway activity is higher in escapees than non-escapees. This new main figure has a dedicated section in the main text.

To test the contribution of MAPK pathway reactivation to the escape phenomenon, we previously co-treated cells with dabrafenib + trametinib and measured the response by time-lapse microscopy and immunofluorescence (revised Fig. 3b-d). We also added a new time-lapse experiment in which we co-treated cells with dabrafenib + SHP099, which blocks RTK signaling arising from dabrafenib-induced ‘paradoxical’ reactivation of the MAPK pathway (revised Supplementary Fig. 2g-h). In both experiments, co-drugging reduced the fraction of A375 escapees from 43% to 3%, clearly showing that MAPK reactivation is an important contributor to the escape phenomenon.

However, we could not fully eliminate the escapee population even with ‘fully’ MAPK-inhibited cells (ie 3% of escapees remain), indicating that other mechanisms contribute to escape in addition to MAPK pathway reactivation. Our scRNA-seq data pointed us to a potential role for ATF4 in supporting escapees, and indeed, siRNA knockdown of ATF4 followed by dabrafenib, trametinib, or dabrafenib + trametinib further reduces the escapee population (revised Fig. 4e). This demonstrates a role for ATF4 stress signaling even when the MAPK pathway is ‘fully’ inhibited.

Also, the authors argue that the “subpopulation of escapees” relies on stress signaling mediated by ATF4. This conclusion is based on the filters that they have set on the gene set enrichment analysis (Fig. 3b, c) to identify the genes that are differentially expressed in “both” dabrafenib-treated escapees vs. non-escapees and in dabrafenib-treated escapees vs. untreated proliferating cells. Based on this analysis, one would expect that untreated proliferating cells or dabrafenib-treated non-escapees should be insensitive to ATF4 knockdown. However, ATF4 siRNA data are only shown for dabrafenib-treated cells but not for untreated cells or for escapees vs. non-escapees (Fig. 3e).

We thank the reviewer for pointing out the necessary siATF4-only control, which was accidentally omitted. This is now included with the relevant statistical testing and shows either no effect (SKMEL28) or slightly improved proliferation (A375) by itself (revised Fig. 4e). That is, ATF4 knockdown hinders proliferation in dabrafenib-treated cells but not untreated cells.

In teasing out the functional role of ATF4, we speculated that ATF4 may be protecting drug-treated cells from an apoptotic fate. Indeed, new experiments show that dabrafenib, trametinib, and co-treated cells transfected with ATF4 siRNA experience a significantly increased level of apoptosis compared to siControl (revised Fig. 4f). These new data further support the notion that cells escaping ‘full’ MAPK pathway inhibition (dabrafenib + trametinib) also rely on ATF4. A section of main text has been added to emphasize this important point.

The prognosis statistical analysis should be improved in order to support the authors' conclusion about the significance of ATF4/mTORC1-associated genes for patient survival. The authors show that only 5 out of 40 genes correlate negatively with patient survival. The question is: how many genes out of any 40 randomly selected genes would correlate negatively with patient survival using the exact same approach? This is important to address, as authors show that 10% and 7.5% genes in the genome have significantly positive and negative hazard ratios. Therefore, for authors' conclusion (about the clinical relevance of ATF4/mTORC1-associated genes) to have statistical significance, a permutation analysis should be performed, based on which p-values should be corrected.

Thank you for your suggestion. In our 38-gene list (excluding the two genes not found in the TCGA datasets), five genes had negative correlation with patient survival (significantly positive hazard ratios; significant log-rank p-values for KM plots; and the median survival of patients with top 25% expression levels was shorter than the patients with bottom 25% expression levels) while none had positive correlation (significantly negative hazard ratios; no requirement on KM plots or median survival). As per these criteria, 5.6% and 7.5% genes in the genome had negative and positive correlation. Therefore, the probability of obtaining a better enrichment (at least 5 genes had negative correlation while none had positive correlation) through random sampling would equal to $\sum_{i=5}^{38} C_{38}^i 0.056^i (1 - 0.056 - 0.075)^{38-i} = 0.004 \ll 0.05$. Therefore, we claimed that the genes with long-term negative correlations on patient survival were indeed significantly enriched in our gene list. This information is now included in the "Prognostic Analysis" section of the Methods.

REVIEWERS' COMMENTS

Reviewer #3 (Remarks to the Author):

The authors have addressed all of the the points raised in my review.